# The role of Pitx2 and Pitx3 in muscle stem cells gives new insights into P38α MAP kinase and redox regulation of muscle regeneration

Aurore L'honoré[1,2†*], Pierre-Henri Commère[3], Elisa Negroni[4], Giorgia Pallafacchina[5], Bertrand Friguet[2], Jacques Drouin[6], Margaret Buckingham[1], Didier Montarras[1*]

[1]Department of Developmental and Stem Cell Biology, CNRS, UMR 3738, Institut Pasteur, Paris, France; [2]Biological Adaptation and Aging-IBPS, CNRS UMR 8256, INSERM ERL U1164, Sorbonne Universités, Université Pierre et Marie Curie, Paris, France; [3]Platform of Cytometry, Institut Pasteur, Paris, France; [4]Center for Research in Myology, Sorbonne Universités, Université Pierre et Marie Curie, Paris, France; [5]NeuroscienceInstitute, Department of Biomedical Sciences, Italian National Research Council, Universityof Padova, Padova, Italy; [6]Laboratory of Molecular Genetics, Institut de Recherches Cliniques de Montréal, Montréal, Canada

**\*For correspondence:**
alhonore@hotmail.com (AL'é);
didier.montarras@pasteur.fr (DM)

**Present address:** †Sorbonne Universités, UPMC Université Paris 06, Biological Adaptation and Aging-IBPS, CNRS UMR 8256, INSERM ERL U1164, Paris, France

**Abstract** Skeletal muscle regeneration depends on satellite cells. After injury these muscle stem cells exit quiescence, proliferate and differentiate to regenerate damaged fibres. We show that this progression is accompanied by metabolic changes leading to increased production of reactive oxygen species (ROS). Using *Pitx2/3* single and double mutant mice that provide genetic models of deregulated redox states, we demonstrate that moderate overproduction of ROS results in premature differentiation of satellite cells while high levels lead to their senescence and regenerative failure. Using the ROS scavenger, N-Acetyl-Cysteine (NAC), in primary cultures we show that a physiological increase in ROS is required for satellite cells to exit the cell cycle and initiate differentiation through the redox activation of p38α MAP kinase. Subjecting cultured satellite cells to transient inhibition of P38α MAP kinase in conjunction with NAC treatment leads to their rapid expansion, with striking improvement of their regenerative potential in grafting experiments.
DOI: https://doi.org/10.7554/eLife.32991.001

## Introduction

In adult mammals, many tissues can undergo repair in response to injury due to the presence of stem cells. In contrast to embryonic stem cells, most adult stem cells are maintained in a quiescent state in specialized niches (*Lander et al., 2012*; *Ema and Suda, 2012*). In response to tissue damage or changes in their microenvironment, they exit quiescence, proliferate and then differentiate in order to regenerate the tissue, or self-renew to reconstitute the stem cell pool. Progression from quiescence to proliferation and differentiation requires metabolic flexibility to adapt to changing energy demands, as exemplified by hematopoietic stem cells which when quiescent mainly rely on glycolysis, but upon activation depend on active mitochondrial biogenesis and oxidative phosphory-lation (*Simsek et al., 2010*; *Takubo et al., 2013*). Such a metabolic switch, which is critical for responding to bio-energetic needs, may also be implicated in the regulation of adult stem cell behaviour, as observed for reactive oxygen species (ROS) that are by-products of oxidative

phosphorylation (*Murphy, 2009*). Although detrimental when in excess, ROS can serve as signalling molecules at physiological levels (*D'Autréaux and Toledano, 2007*; *Sena and Chandel, 2012*), and their regulation is now emerging as an important facet of stem cell biology (*Bigarella et al., 2014*; *Khacho and Slack, 2017*; *Le Moal et al., 2017*). Thus increased ROS production when quiescent hematopoietic stem cells switch from glycolysis to oxidative phosphorylation has been shown to be critical for priming their activation (*Jung et al., 2013*). In addition to enhancing stem cell proliferation, moderate levels of ROS can also affect cell fate decisions (*Khacho and Slack, 2017*). However, it is still unclear how ROS levels in adult stem cells can modulate the critical balance between proliferation and differentiation in an in vivo context and whether this impacts tissue homeostasis and regeneration.

To address this issue we have used the model of adult muscle stem cells, known as satellite cells, which progress from quiescence to proliferation and differentiation during skeletal muscle regeneration. While quiescent satellite cells have a low redox state, we observed a marked increase in ROS levels in activated cells, with a peak occurring prior to the onset of muscle differentiation. To investigate the effect of redox regulation on satellite cell behaviour, we employed conditional mouse mutants for *Pitx2* and *Pitx3* genes (*Gage et al., 1999a*), encoding homeodomain transcription factors that we previously identified as critical regulators of cell redox state during foetal myogenesis (*L'honoré et al., 2014*). We found that the intracellular ROS level is a critical regulator of satellite cell behaviour, acting through p38α MAP kinase activity. While the moderate overproduction of ROS observed in the single *Pitx3* mutant results in the premature differentiation of satellite cells, excessive ROS levels seen in double *Pitx2/3* mutants lead to impaired skeletal muscle regeneration due to accumulation of DNA damage and senescence of satellite cells. Reduction of ROS levels by the antioxidant N-Acetyl-Cysteine (*Richards et al., 2011*), together with inhibition of P38α MAP kinase signalling (*Segalés et al., 2016*), leads to robust expansion of satellite cells in culture. Satellite cells cultured under these conditions show high in vivo expansion and regenerative potential upon grafting, with implications for muscle cell therapy.

## Results

### Increased ROS and mitochondrial biogenesis mark the progression of satellite cells towards terminal differentiation

To investigate the regulation of mitochondrial metabolism in quiescent and committed myoblasts, we first performed a transcriptome analysis with Pax3(GFP)-positive satellite cells (*Pallafacchina et al., 2010*) purified by flow cytometry from adult $Pax3^{GFP/+}$ (Adult), postnatal day 7 $Pax3^{GFP/+}$ (P7) and adult dystrophic $mdx/mdx:Pax3^{GFP/+}$ ($mdx$) mice (*Figure 1A*). Compared with quiescent satellite cells from adult muscle, activated satellite cells present during post-natal development, or in $mdx$ muscles (*Pallafacchina et al., 2010*), which in the absence of Dystrophin undergo chronic regeneration, showed up-regulation of genes implicated in fatty acid metabolism and in oxidative phosphorylation, including regulators of mitochondrial biogenesis and function (*Figure 1A*, *Figure 1—figure supplement 1A*). Such activated cells, marked by the onset of *Myogenin* transcription (*Figure 1—figure supplement 1B*), display increased levels of ROS (*Figure 1—figure supplement 1C,D*), showing that activation is accompanied by metabolic changes involving increase in both mitochondrial activity and ROS production. We then investigated mitochondrial activity during the transition from proliferation to differentiation. Satellite cells were purified by flow cytometry from *pectoralis*, diaphragm and abdominal muscles of adult $Pax3^{GFP/+}$ mice and both respiration and glycolysis were measured by Seahorse analysis after different days of culture (D2-D4) (*Figure 1—figure supplement 1E–G*). While proliferative satellite cells (D2) predominantly display glycolytic metabolism, the contribution of oxidative phosphorylation increases at day 3 (D3, *Figure 1—figure supplement 1E–G*), a time that precedes differentiation by one day, as evidenced by *Myogenin* expression (*Figure 1—figure supplement 1J*). To investigate the kinetics of mitochondrial function and of ROS production in vivo during muscle regeneration, satellite cells were then purified by flow cytometry from *pectoralis* muscles of adult $Pax3^{GFP/+}$ mice at different days (D0 to D7) after notexin-induced injury (*Figure 1B*). While quiescent satellite cells are characterized by a low mitochondrial mass (*Latil et al., 2012*; *Tang and Rando, 2014*), we observed an increase in mitochondrial mass occurring two days after injury and continuing thereafter (*Figure 1C,D*). This increase precedes the

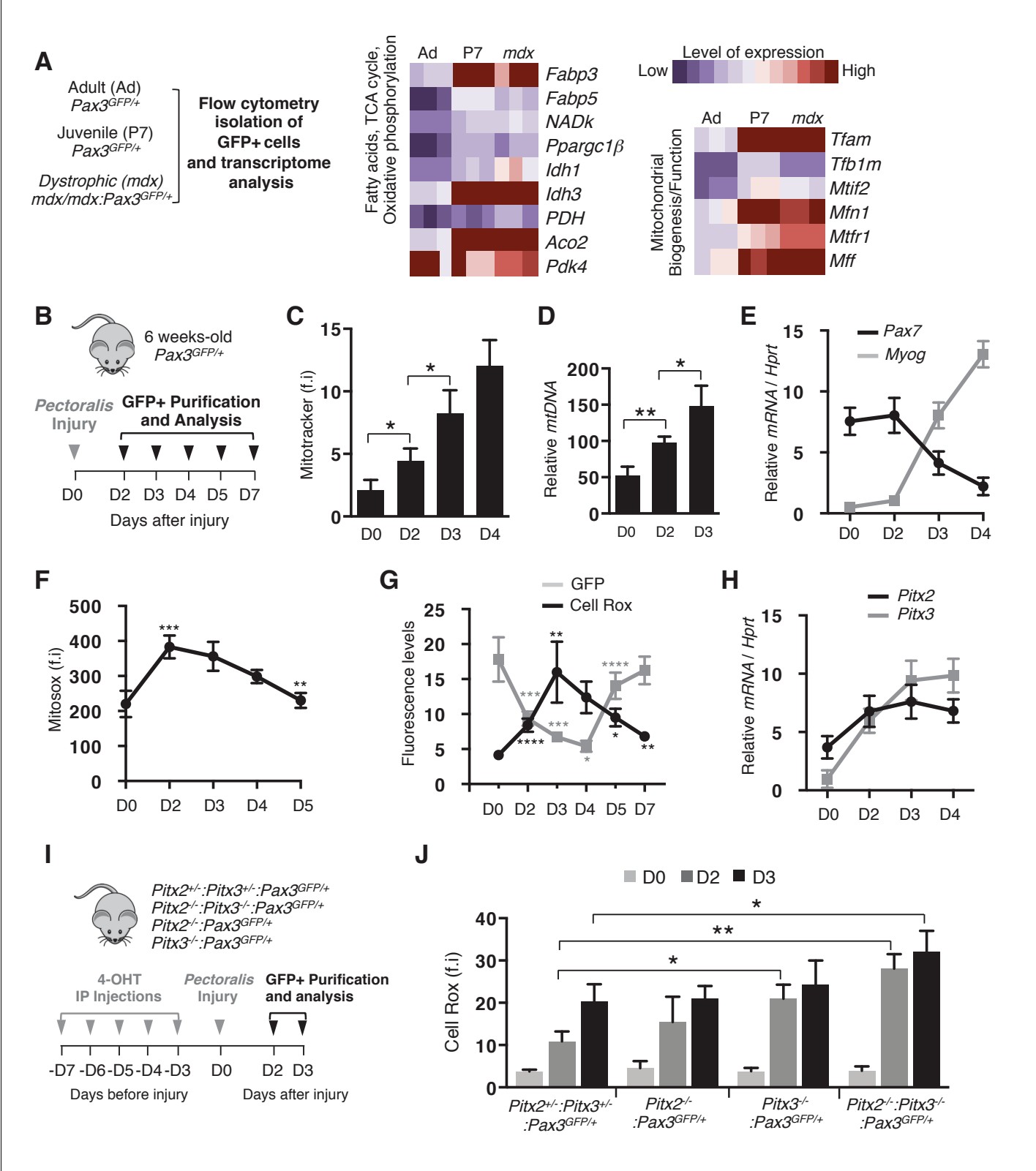

**Figure 1.** Activation of muscle stem cells during regeneration is accompanied by changes in their redox state. (**A**) Quiescent (adult) and activated (P7 and *mdx*) satellite cells were purified respectively from adult *Pax3^GFP/+^* (Ad, 98% quiescent), postnatal day 7 *Pax3^GFP/+^* (P7, 80% activated) and adult dystrophic *mdx:Pax3^GFP/+^* (*mdx*, 30% activated) mice. In each case, *pectoralis*, abdominal and diaphragm muscles were used for isolation by flow cytometry of Pax3(GFP)-positive satellite cells. After purification, RNA was isolated from Pax3(GFP)-positive cells for transcriptome analysis

*Figure 1 continued on next page*

Figure 1 continued

(**Pallafacchina et al., 2010**). (**B**) Experimental scheme used for the purification of quiescent and activated satellite cells during regeneration. (**C–H**) GFP-positive cells were isolated from the *pectoralis* muscles of adult $Pax3^{GFP/+}$ mice at different days (**D**) after notexin injection or from control muscles at D0. (**C**) GFP-positive cells isolated at D0 (quiescent) and D2 to D4 after notexin injury (activated) were immediately incubated with Mitotracker probe and analysed by flow cytometry to measure fluorescence intensity (f.i). (**D**) DNA samples prepared from Pax3(GFP)-positive cells were analysed by qPCR for the level of the mitochondrial *COI* gene relative to the level of the nuclear encoded *NDUFV1* gene, expressed as a log ratio. (**E**) RNA samples prepared from Pax3(GFP)-positive cells were analysed by qPCR for transcripts of *Pax7* and *Myogenin* relative to the level of *Hprt* transcripts, expressed as a log ratio. (**F, G**) Immediately after purification at D0-D5 (**F**) or D0-D7 (**G**), GFP-positive cells were incubated with Mitosox (**F**) or Cell Rox (**G**) probes and the relative fluorescence intensity (f.i) measured by flow cytometry. (**H**) RNA samples as in (**E**) were analysed by qPCR for *Pitx2* and *Pitx3* transcripts. (**I**) Double *Pitx2/3* mutant $Pitx2^{flox/flox}:Pitx3^{flox/-}:Pax3^{GFP/+}:R26R^{Cre-ERT2/+}$, single *Pitx2* mutant $Pitx2^{flox/flox}:Pax3^{GFP/+}:R26R^{Cre-ERT2/+}$, single *Pitx3* mutant $Pitx3^{-/-}:Pax3^{GFP/+}$, and control $Pitx2^{flox/+}:Pitx3^{+/-}:Pax3^{GFP/+}:R26R^{Cre-ERT2/+}$ adult mice were obtained by 4-hydroxytamoxifen (4-OHT) intra-peritoneal (IP) injection on 5 consecutive days (**D**) and subjected to muscle injury by notexin injection into the *pectoralis* muscle (D0). (**J**) GFP-positive cells isolated as in (**B**) at D0, 2 and 3 after injury were incubated with Cell Rox probe and the relative fluorescence intensity (f.i.) measured by flow cytometry. Error bars (**C–H, J**) represent the mean $\pm$ s.d with n $\geq$ 6 independent animals (**C, E–H, J**) or 4 independent animals (**D**). Error bars represent the mean $\pm$ s.d, with *p<0.05, **p<0.01, ***p<0.001, ****p<0.001. Please see *Figure 1—figure supplement 1* for additional data.

DOI: https://doi.org/10.7554/eLife.32991.002

The following source data and figure supplements are available for figure 1:

**Source data 1.** Numerical data used to generate *Figure 1*.
DOI: https://doi.org/10.7554/eLife.32991.005
**Figure supplement 1.** Activation of muscle stem cells is accompanied by mitochondrial activity.
DOI: https://doi.org/10.7554/eLife.32991.003
**Figure supplement 1—source data 1.** Numerical data used to generate *Figure 1—figure supplement 1*.
DOI: https://doi.org/10.7554/eLife.32991.004

onset of differentiation, marked by *Myogenin* expression and down-regulation of *Pax7* (*Figure 1E*). Strikingly, quantification of total $H_2O_2$ production and mitochondrial ROS production revealed a marked increase at two days after activation (*Figure 1F*, *Figure 1—figure supplement 1H*), a time when satellite cells initiate differentiation (*Figure 1E*). Analysis of cytoplasmic ROS levels showed an increase at day 2 after injury that reaches a peak at day 3 (*Figure 1G*), when the level of GFP, reflecting *Pax3* transcription, is low, as also seen for *Pax7* (*Figure 1E*). Subsequent increase in GFP fluorescence (*Figure 1G*) corresponds to later self-renewal of the Pax3-positive satellite cell pool.

As these results clearly indicate that intracellular ROS content reaches a peak before the onset of satellite cell differentiation, we then examined whether it is required to drive differentiation. To address this issue, and since we had previously identified *Pitx2/3* genes as essential regulators of the redox state in foetal muscle precursor cells (*L'honoré et al., 2014*), we employed conditional *Pitx2/3* mutant mice. Both *Pitx2* and *Pitx3* transcripts and their corresponding proteins are up-regulated in activated satellite cells, marked by MyoD expression and then by differentiation markers Myogenin and Troponin T, in vivo after injury (*Figure 1H*, *Figure 1—figure supplement 1I*), and ex vivo in cultured satellite cells (*Knopp et al., 2013*) (*Figure 1—figure supplement 1J–L*).

Conditional ablation of the floxed-*Pitx2* gene (*Gage et al., 1999b*) alone, or in combination with the constitutive *Pitx3* mutant (*L'honoré et al., 2014*) was carried out by five intra-peritoneal injections of 4-hydroxy-tamoxifen (4-OHT). Pax3(GFP)-positive cells were then purified by flow cytometry from the *pectoralis* muscles of $Pitx2^{flox/+}:Pitx3^{+/-}:Pax3^{GFP/+}:R26R^{Cre-ERT2/+}$ controls, $Pitx3^{-/-}:Pax3^{GFP/+}$ and $Pitx2^{flox/flox}:R26R^{Cre-ERT2/+}:Pax3^{GFP/+}$ single mutants and $Pitx2^{flox/flox}:Pitx3^{flox/-}:R26R^{Cre-ERT2/+}:Pax3^{GFP/+}$ double mutants, in the absence of injury (D0) or 2 and 3 days (D2, D3) after notexin injection (*Figure 1I*). As previously shown (*Figure 1F,G*), ROS levels increase markedly from day 2 after injury in control satellite cells isolated from $Pitx2^{flox/+}:Pitx3^{+/-}:Pax3^{GFP/+}:R26R^{Cre-ERT2/+}$ mice (*Figure 1J*). While *Pitx2* mutant satellite cells display a similar redox state as control cells during regeneration, we observed abnormal moderate or high increases in ROS at days 2 and 3 after injury in satellite cells from single *Pitx3* or double *Pitx2/3* mutant mice, respectively (*Figure 1J*). Comparison of ROS levels in Pax3(GFP) satellite cells isolated from double heterozygotes versus wild-type mice after notexin injury gave similar results (*Figure 1—figure supplement 1M*). We therefore systematically employed the double heterozygote as a control.

# Increased ROS production is critical for satellite cell progression from proliferation to differentiation through redox activation of p38α MAP kinase

To test the consequences of a moderate ROS increase on activated satellite cell behaviour, we used *Pitx3*$^{-/-}$ mice. At birth, *Pitx3* mutants were indistinguishable from controls, but then displayed a progressive growth defect (*Figure 2—figure supplement 1A*), with reduced muscle mass and a decrease in the myofibre cross-sectional area (CSA) (*Figure 2—figure supplement 1B,C*), a phenotype that persists in adults (*Figure 2A,B*, *Figure 2—figure supplement 1D*). To investigate the behaviour of *Pitx3* mutant satellite cells during muscle regeneration, we injured the *Tibialis Anterior (TA)* muscle of *Pitx3*$^{-/-}$ mutant and *Pitx3*$^{+/-}$ animals by cardiotoxin injection. Three weeks after injury, regenerated fibres of control mice have a similar size to that of uninjured fibres, whereas mutant muscles exhibit a decreased CSA (*Figure 2A,B*). This phenotype, which is correlated with a deficit in centrally located myonuclei in regenerating mutant fibres compared to controls (*Figure 2C*), may reflect abnormal behaviour of *Pitx3* mutant satellite cells. We therefore examined the kinetics of their differentiation during regeneration. While in control animals, most satellite cells are still proliferative 3 days after injury, *Pitx3* mutant cells show a marked decrease in the number of Pax7/EdU double positive cells (*Figure 2D,E*), together with a two fold increase in the number of differentiating cells marked by Myogenin and embryonic-Myosin Heavy Chain (Emb-MyHC) (*Figure 2D,F*). These results indicate that *Pitx3*-deficient satellite cells undergo premature differentiation, a feature that could account for the muscle regeneration defect observed in *Pitx3*$^{-/-}$ mice.

To challenge their regenerative potential, *Pitx3* mutant and control mice were subjected to three rounds of *TA* muscle injury by cardiotoxin injection (*Figure 2G*). After the last round, muscle sections from control animals displayed a normal pattern of regenerating centro-nucleated fibres (*Figure 2H–J*). In contrast, *Pitx3* mutant muscles exhibited impaired regeneration marked by fibrosis (*Figure 2H*), a severe reduction in the regenerated fibre CSA (*Figure 2I*, *Figure 2—figure supplement 1D,E*), and by depletion of the satellite cell reservoir (*Figure 2J*) with a twofold reduction in the number of Pax7-positive cells. Similarly, adult *Pitx3*$^{-/-}$:*mdx/mdx* mice display regeneration defects with muscle fibre hypoplasia and fibrosis, (*Figure 2—figure supplement 1F* and data not shown).

To evaluate the contribution of altered redox state to the premature differentiation of *Pitx3* mutant satellite cells, Pax3(GFP)-positive cells were purified by flow cytometry from muscles of control *Pitx3*$^{+/-}$:*Pax3*$^{GFP/+}$ and mutant *Pitx3*$^{-/-}$:*Pax3*$^{GFP/+}$ adult mice and their behaviour analysed in culture in the presence or absence of the ROS scavengers N-Acetyl-Cysteine (NAC) (*Richards et al., 2011*) or Trolox (*Le Moal et al., 2017*). As expected, while the absence of Pitx3 does not affect satellite cell activation marked by MyoD expression (data not shown), it leads to their premature differentiation. This phenotype is illustrated by a significant increase in the number of Myogenin- and Troponin T-positive cells at days 3 and 4 in the mutant population compared to controls (*Figure 3A*, *Figure 3—figure supplement 1A–E*). In keeping with the in vivo results, *Pitx3* mutant cells exhibit higher ROS levels than control cells at day 3 (*Figure 3—figure supplement 1F*), together with reduced proliferation potential and premature cell cycle exit, shown by clonal analysis (*Figure 3B*) and by EdU and p21 staining (*Figure 3C*). The decrease in ROS driven by the use of the antioxidants NAC or Trolox (*Figure 3—figure supplement 1F*) efficiently prevents premature differentiation of *Pitx3* mutant cells (*Figure 3A*), and results in a rescue of their proliferation potential (*Figure 3B,C*). Culturing the *Pitx3* mutant cells in 3% $O_2$ does not rescue their phenotype, while under these normoxic conditions the response to NAC is similar to that seen in 20% $O_2$ (*Figure 3—figure supplement 1G*).

We then investigated the mechanism by which ROS regulate the onset of differentiation and focussed on the redox sensor p38α MAP kinase (*Ito et al., 2006*; *Brien et al., 2013*; *Segalés et al., 2016*). Measurement of p38α MAP kinase phosphorylation demonstrated its premature activationin *Pitx3* mutant cells at day 3 of culture (*Figure 3D*), concomitantly with Myogenin expression (*Figure 3A*). In addition, treatment of *Pitx3* mutant cells with either the p38α MAP kinase chemical inhibitor SB203580 (*Bain et al., 2007*) (*Figure 3E*), siRNA directed against p38α MAP kinase (*Figure 3F*), or NAC (*Figure 3A*) prevented premature Myogenin expression and p38α MAP kinase activation (*Figure 3G–I*). Altogether, these results indicate that the increased ROS level observed in activated Pitx3-deficient satellite cells leads to their premature differentiation through activation of

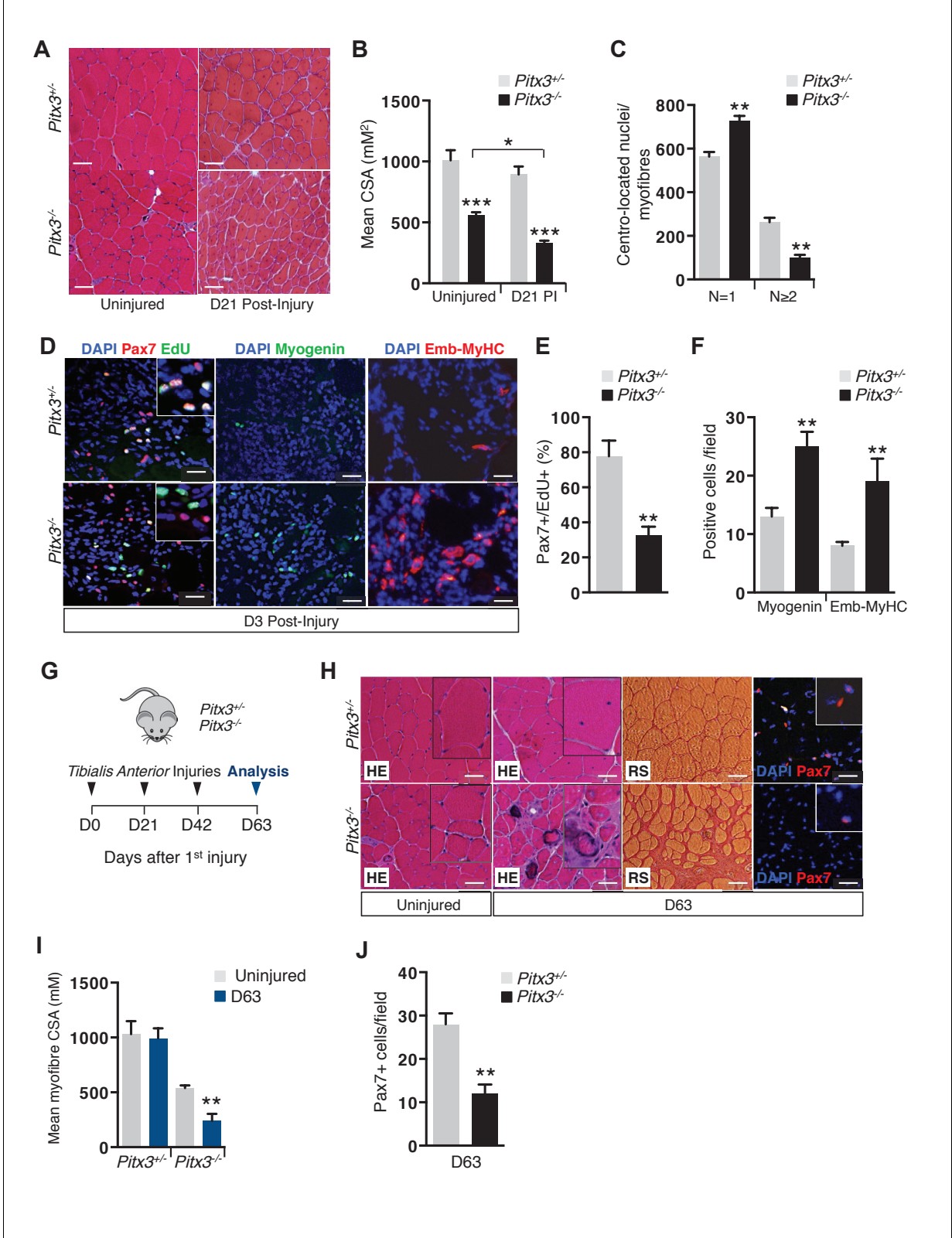

**Figure 2.** Absence of Pitx3 in adult muscle leads to hypoplasia of regenerated fibres by defective amplification and premature differentiation of mutant satellite cells. *Mutant Pitx3⁻/⁻ and control Pitx3⁺/⁻ adult mice were subjected to muscle injury by cardiotoxin injection into the Tibialis Anterior (TA)* muscle. (**A**) Twenty-one days (D21) after injection, injured and contralateral uninjured *TA* muscles were dissected and cryo-sections were analysed by hematoxylin-eosin (HE) staining. (**B**) The cross-sectional area (CSA) of control and mutant fibres was measured in the absence of injury and 21 days (D21)
*Figure 2 continued on next page*

*Figure 2 continued*

post injury (PI) and represented as a mean value. (**C**) The number of fibres with one or two and more centro-located nuclei was counted on transverse sections of control and mutant muscle, and expressed per section. (**D**) Three days after injury, control and mutant animals were treated with EdU for 5 hr and muscles were dissected. Muscle cryo-sections were stained with DAPI (blue) and analysed by immunofluorescence with Pax7 (red), Myogenin (green) and Embryonic Myosin Heavy Chain (Emb-MyHC, red) antibodies and by the EdU reaction (green). (**E**) Pax7/EdU double positive-cells expressed as a percentage of Pax7-positive cells and (**F**) Myogenin-positive and Emb-MyHC-positive cells were counted per field. (**G–J**) Mutant *Pitx3*[-/-] and control *Pitx3*[+/-] mice were subjected to three successive muscle injuries (D0, D21 and D42) by cardiotoxin injection into the *TA* muscle. Three weeks after the third injection (D63), injured and contralateral uninjured muscles were dissected. (**H**) Cryo-sections were analysed by hematoxylin-eosin (HE) and red sirius (RS) staining and by immunofluorescence with a Pax7 (red) antibody on DAPI stained sections. (**I**) CSA of control and mutant fibres was measured in regenerated and contralateral uninjured TA muscles, and is represented as a mean value. (**J**) Pax7 positive-cells were counted and quantified per field. The experiments were performed with n ≥ 3 animals, and a representative image is shown. (B, C, E, F, I, J), error bars represent the mean ±s.d, with **p<0.01, ***p<0.001. (A, H) Scale bar, 100 μm, (D) Scale bar, 50 μm. Please see *Figure 2—figure supplement 1* for additional data.

DOI: https://doi.org/10.7554/eLife.32991.006

The following source data and figure supplements are available for figure 2:

**Source data 1.** Numerical data used to generate *Figure 2*.

DOI: https://doi.org/10.7554/eLife.32991.009

**Figure supplement 1.** *Pitx3* deletion leads to defective skeletal muscle growth and to aggravation of the *mdx* dystrophic phenotype.

DOI: https://doi.org/10.7554/eLife.32991.007

**Figure supplement 1—source data 1.** Numerical data used to generate *Figure 2—figure supplement 1*.

DOI: https://doi.org/10.7554/eLife.32991.008

the p38α MAP kinase pathway. We also verified ERK, JUNK, NFκB and ATM signalling pathways, but did not observe any significant change in the ratio of phosphorylated to total forms of ERK and JUNK kinases (*Figure 3—figure supplement 1H,I*), while no phosphorylation of p65 (NFκB) nor ATM could be detected (data not shown) in the *Pitx3* mutant cells after 3 days of culture. Since it is possible that Pitx3 also controls ROS levels in response to cytokine stimulation, we treated control and mutant satellite cells with 0.5 nM TNFα. As shown in *Figure 3—figure supplement 1J*, stimulation by TNFα leads to a similar increase in ROS in both control and *Pitx3* mutant cells, indicating that ROS production following cytokine stimulation does not depend on Pitx3.

To further document the role of ROS in satellite cell differentiation, we manipulated ROS levels in cultured satellite cells purified from control *Pax3*[GFP/+] adult mice (*Figure 3J–N*, *Figure 3—figure supplement 1J–O*). In view of the increase in ROS levels after TNFα stimulation (*Figure 3—figure supplement 1J*), we first investigated their consequences on satellite cell proliferation and differentiation. As shown in *Figure 3J*, addition of 0.5 nM TNFα leads to reduced proliferation together with premature differentiation. These effects are prevented by the presence of NAC, showing they result from ROS increase.

Control Pax3(GFP)-positive satellite cells were transfected with siRNA directed against transcripts encoding the mitochondrial uncoupling factor, *Ucp2* (*Mailloux and Harper, 2011*) and analysed after 3 days of culture for mitochondrial content, ROS levels, and for Myogenin expression. As expected, inhibition of mitochondrial uncoupling (*Figure 3—figure supplement 2A*) did not affect mitochondrial content (*Figure 3—figure supplement 2B*) but led to an increase in total ROS levels (*Figure 3K*) and to premature satellite cell differentiation as shown by an increased number of phospho-p38α- (pp38α) and Myogenin-positive cells (*Figure 3L*, *Figure 3—figure supplement 1M*). In contrast, repression of mitochondrial biogenesis by silencing of *Tfam* (*Gleyzer et al., 2005*) using *siRNA* led to a decrease in ROS levels together with delayed differentiation (*Figure 3—figure supplement 2D and E*). We then investigated whether NADPH oxidases, enzymes present at the cell membrane and in the endoplasmic reticulum, could also contribute to ROS production in activated satellite cells. qRT-PCR analyses indicated that *Nox2* and *Nox4* expression increases in the days (D2-D4) following injury (*Figure 3—figure supplement 2F*). Inhibition of both enzymes by Apocynin (*Petrônio et al., 2013*) leads to a decrease in ROS levels (*Figure 3M*), together with an inhibition of differentiation and a maintenance of proliferation, as shown at day 4 by the reduced number of pp38α- and Myogenin-positive cells, and the increased fraction of EdU-incorporating cells, compared to controls (*Figure 3N*, *Figure 3—figure supplement 2G*).

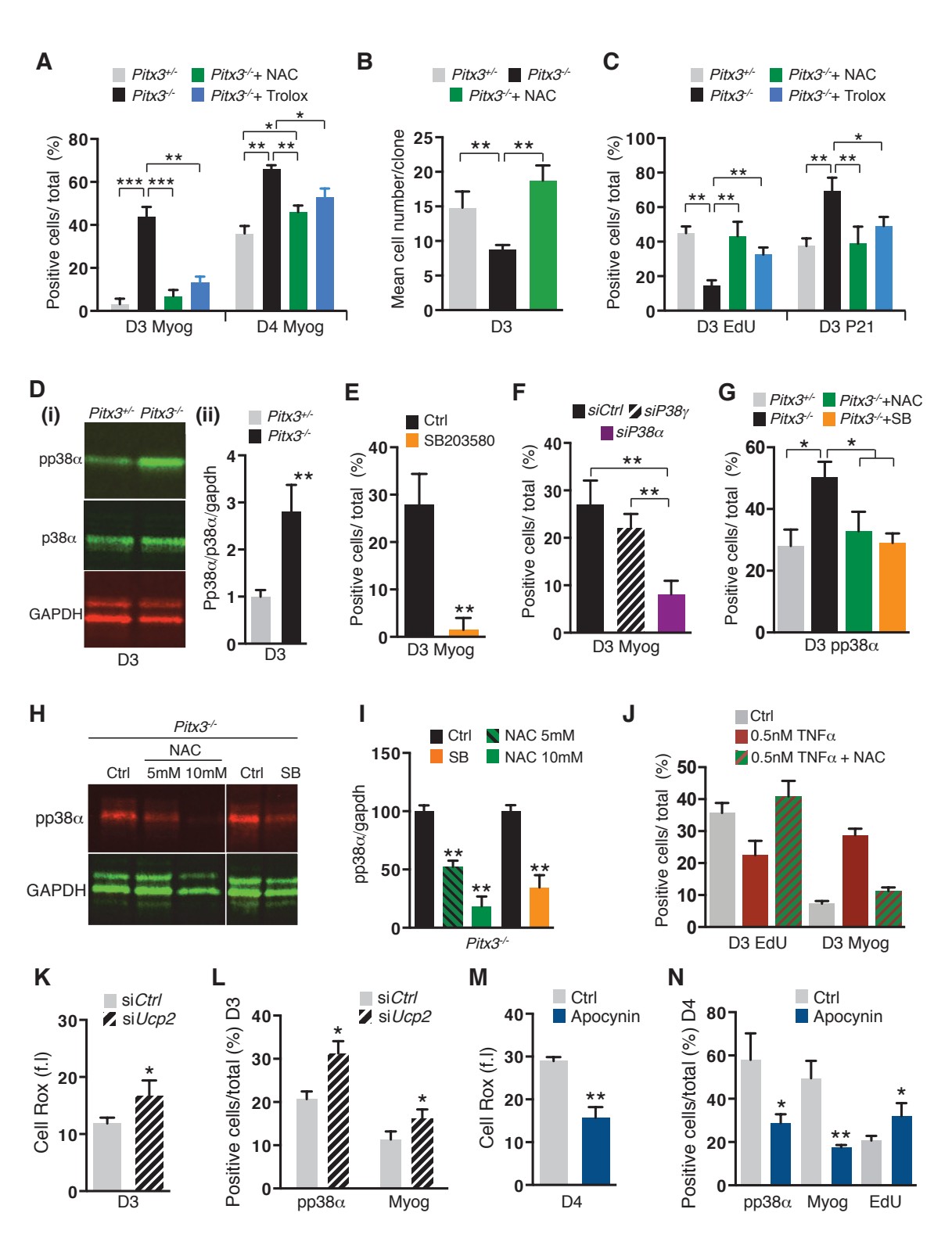

**Figure 3.** Premature differentiation of *Pitx3* mutant satellite cells is mediated by activation of p38α MAP kinase and can be prevented by ROS scavenging or p38α MAP kinase inhibition. In all experiments, GFP-positive cells were isolated by flow cytometry from the *pectoralis*, abdominal and diaphragm muscles of mutant *Pitx3⁻/⁻:Pax3^GFP/+* and control *Pitx3⁺/⁻:Pax3^GFP/+* adult mice. (A–C) Cells were cultured in the presence or absence of 10 mM N-Acetyl Cysteine (NAC) or of 10 µM Trolox. (A) After 3 and 4 days of culture, control and mutant cells were stained with DAPI and by

*Figure 3 continued on next page*

*Figure 3 continued*

immunofluorescence using a Myogenin antibody. Myogenin-positive cells (Myog) were counted and quantified as the percentage (%) of total cells. (B) After purification, control and mutant satellite cells were used for clonal analysis. Their proliferation capacity is expressed as the mean cell number per clone after 3 days of culture in the presence or absence of 10 mM NAC. (C) After 3 days of culture in the presence or absence of 10 mM NAC or of 10 μM Trolox, control and mutant GFP-positive cells were incubated with EdU. Cells were processed for detection of EdU incorporation and p21 expression by immunofluorescence using a p21 antibody. p21 and EdU-positive cells were counted and quantified as the percentage (%) of total cells. (D) GFP-positive cells isolated from mutant *Pitx3$^{-/-}$:Pax3$^{GFP/+}$* mice and control *Pitx3$^{+/-}$:Pax3$^{GFP/+}$* adult mice were cultured for 3 days and proteins were extracted. Western-blot analysis indicated the level of pp38α, p38α and GADPH proteins (i). Quantification of western blots based on densitometry fluorescence scans for pp38α, p38α and GAPDH (ii). (E, F) GFP-positive cells isolated from *Pitx3$^{-/-}$:Pax3$^{GFP/+}$* mice were purified and cultured. (E) Mutant *Pitx3$^{-/-}$:Pax3$^{GFP/+}$* cells cultured for 3 days in the presence or absence of SB203580 were analysed by immunofluorescence using a Myogenin antibody. Myogenin-positive cells (Myog) were counted, and quantified as the percentage (%) of total cells. (F) Mutant *Pitx3$^{-/-}$:Pax3$^{GFP}$* cells transfected with siRNA directed against *p38α* and *p38γ* transcripts or siControl were analysed after 3 days of culture by immunofluorescence using a Myogenin antibody. Myogenin-positive cells (Myog) were counted, and quantified as the percentage (%) of total cells. (G) Control *Pitx3$^{+/-}$:Pax3$^{GFP/+}$* and mutant *Pitx3$^{-/-}$:Pax3$^{GFP/+}$* cells cultured for 3 days in the presence or absence of SB203580 or NAC were analysed by immunofluorescence using a pp38α antibody. pp38α-positive cells were counted, and quantified as the percentage (%) of total cells. (H) Mutant *Pitx3$^{-/-}$:Pax3$^{GFP/+}$* cells were cultured in the presence or absence of SB203580 or 5 and 10 mM NAC. After 3 days of culture, proteins were extracted from treated and untreated control cells and processed for western-blot analyses, using antibodies directed against pp38α or GAPDH. (I) Quantification of western-blot analyses as in (D) based on densitometry fluorescence scans indicates the levels of pp38α relative to the GAPDH protein. (J–N) GFP-positive cells isolated by flow cytometry from the *pectoralis*, abdominal and diaphragm muscles of *Pax3$^{GFP/+}$* mice were cultured. (J) 48 hr after culture, GFP-positive cells were treated with 0.5 nM TNFα with and without 10 mM NAC. After 24 hr of treatment, cells were incubated with EdU and processed for detection of EdU incorporation by fluorescence and of myogenin expression by immunofluorescence. Myogenin and EdU-positive cells were counted and quantified as the percentage (%) of total cells. (K, L) GFP-positive cells were transfected 24 hr after plating with siRNA directed against *Ucp2* transcripts, or with siControl (Ctrl). Two days after transfection, cells were processed for quantification of reactive oxygen species (ROS) with the Cell Rox probe (f.i. fluorescence intensity) (K) or for Myogenin (Myog) and phospho-p38α (pp38α) expression by immunofluorescence (L). pp38α- and Myogenin-positive cells were counted, and quantified as the percentage (%) of total cells (L). (M,N) GFP-positive cells were cultured in the presence or absence of Apocynin, the inhibitor of NADPH oxidases. (M) After 4 days in culture, cells were processed for quantification of ROS as in (K), or incubated with EdU and processed for EdU detection by fluorescence, and for Myogenin and pp38α expression by immunofluorescence. (N) pp38α-, Myogenin- and EdU-positive cells were counted and quantified as the percentage (%) of total cells. (A–G, I–N) The experiments were performed with n = 3 animals for each condition. Error bars represent the mean ±s.d, with *p<0.05, **p<0.01, ***p<0.001. Please see *Figure 3—figure supplement 1* for additional data.
DOI: https://doi.org/10.7554/eLife.32991.010

The following source data and figure supplements are available for figure 3:

**Source data 1.** Numerical data used to generate *Figure 3*.
DOI: https://doi.org/10.7554/eLife.32991.015

**Figure supplement 1.** *Pitx3* mutant satellite cells exhibit premature differentiation ex vivo.
DOI: https://doi.org/10.7554/eLife.32991.011

**Figure supplement 1—source data 1.** Numerical data used to generate *Figure 3—figure supplement 1*.
DOI: https://doi.org/10.7554/eLife.32991.012

**Figure supplement 2.**
DOI: https://doi.org/10.7554/eLife.32991.013

**Figure supplement 2—source data 1.** Numerical data used to generate *Figure 3—figure supplement 2*.
DOI: https://doi.org/10.7554/eLife.32991.014

## Excessive ROS levels lead to satellite cell senescence during muscle regeneration

We then investigated the consequences of excessive ROS levels on satellite cell function in regeneration experiments with *Pitx2$^{flox/flox}$:Pitx3$^{flox/-}$:Pax7$^{Cre-ERT2/+}$* double mutants and *Pitx2$^{flox/+}$:Pitx3$^{+/-}$:Pax7$^{Cre-ERT2/+}$* controls (*Figure 4A*). Three weeks after injury, *Pitx2/3* double mutant mice show severely impaired muscle regeneration with very small fibres, signs of fibrosis, and numerous cells undergoing senescence, in comparison to sections of controls, which display the usual pattern of regenerating centro-nucleated fibres (*Figure 4B*). This phenotype was also seen in the absence of Pitx2/3 in the muscles of *mdx* mice, where chronic muscle regeneration takes place in the absence of Dystrophin (*Figure 4C*, *Figure 4—figure supplement 1A*). Muscle sections of *Pitx2$^{flox/+}$:Pitx3$^{+/-}$:mdx/mdx:Pax7$^{Cre-ERT2/+}$* control mice show the expected pattern of on-going regeneration characteristic of *mdx* mice with centro-nucleated fibres (*Figure 4—figure supplement 1A*). In contrast, *Pitx2$^{flox/flox}$:Pitx3$^{flox/-}$:mdx/mdx:Pax7$^{Cre-ERT2/+}$* mice, in which both *Pitx2/3* genes are specifically invalidated in satellite cells, show markedly impaired regeneration with numerous fibrotic areas and the presence of three times more cells undergoing senescence than in control mice (*Figure 4D*,

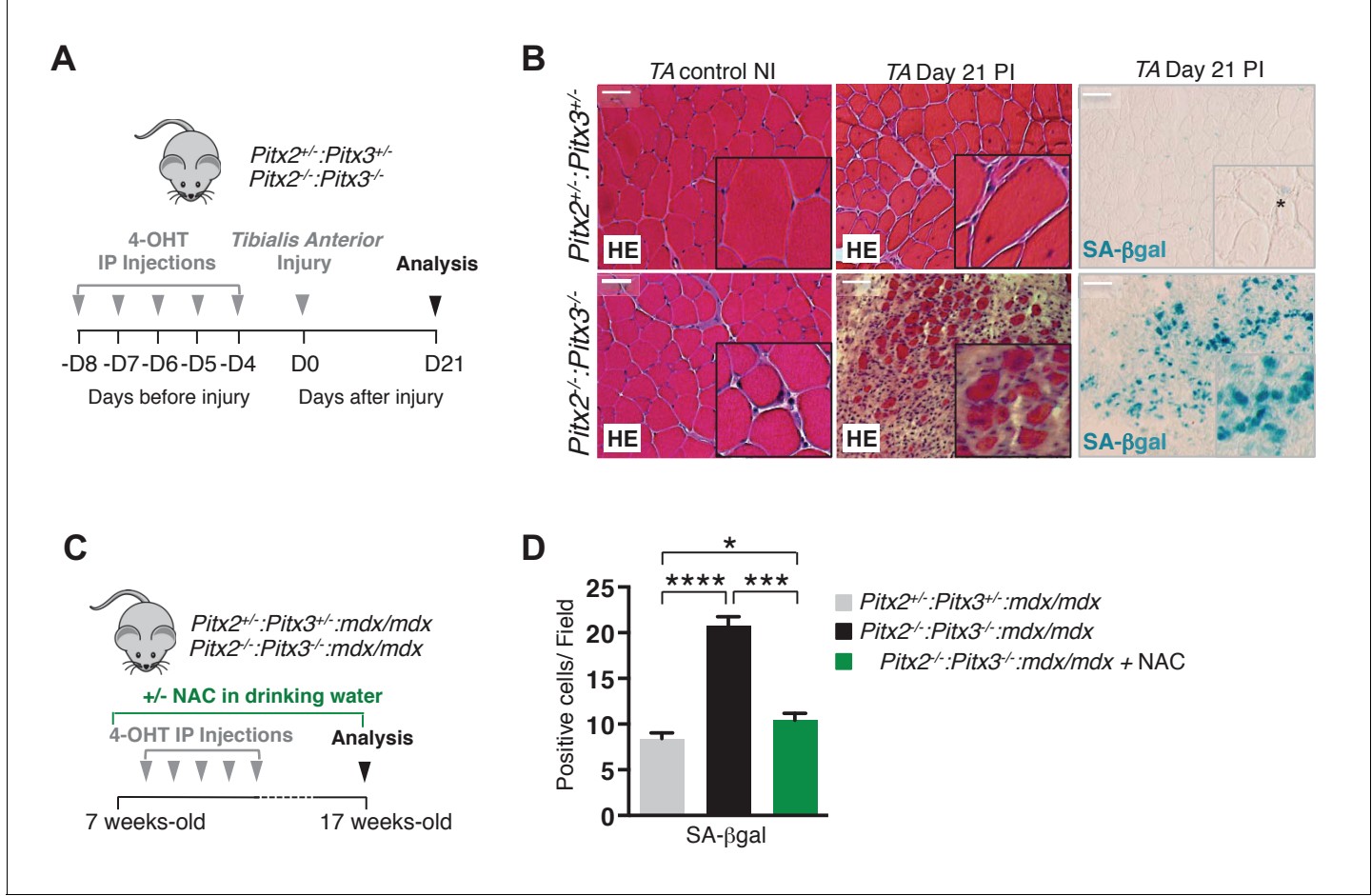

**Figure 4.** Mutation of *Pitx2/3* in adult muscle satellite cells leads to impairment of muscle regeneration by deregulation of their redox state. (A) Double mutant *Pitx2^flox/flox^:Pitx3^flox/-^:Pax7^Cre-ERT2/+^* and control *Pitx2^flox/+^:Pitx3^+/-^:Pax7^Cre- ERT2/+^* adult mice were obtained by 4-hydroxytamoxifen (4-OHT) intra-peritoneal (IP) injection on 5 consecutive days (D) and subjected to muscle injury by cardiotoxin injection into the *Tibialis Anterior* (TA) muscle. (B) Three weeks (D21) post injury (PI), injured and contralateral uninjured *TA* muscles were dissected. Muscle cryo-sections were analysed by hematoxylin-eosin (HE) and senescence-associated−β-galactosidase (SA-βgal) staining. Experiments were performed with n = 4 animals for each condition and representative images are shown, PI (Post Injury). Scale bars, 100 μm. (C) Double mutant *Pitx2^flox/flox^:Pitx3^flox/-^:Pax7^Cre-ERT2/+^:mdx* and control *Pitx2^flox/+^: Pitx3^+/-^: Pax7^Cre-ERT2/+^:mdx* adult mice obtained as in (A) were maintained during 10 weeks under standard life conditions. From 1 week before the start of the experiment, half of the double mutant *Pitx2^flox/flox^:Pitx3^flox/-^:Pax7^Cre-ERT2/+^:mdx* mice were treated with N-Acetyl-Cysteine (NAC) in the drinking water. (D) After 10 weeks, all mice were sacrificed and diaphragm muscles were dissected and analysed by SA-βgal staining. The number of SA-βgal positive cells per field was counted in diaphragm sections of control and double mutant animals. (D) Experiments were performed with n ≥ 5 animals for each condition and representative images are shown (B). Error bars represent the mean ± s.d, with *p<0.05, ***p<0.001, ****p<0.001. Please see *Figure 4—figure supplement 1* for additional data.

DOI: https://doi.org/10.7554/eLife.32991.016

The following source data and figure supplement are available for figure 4:

**Source data 1.** Numerical data used to generate *Figure 4*.
DOI: https://doi.org/10.7554/eLife.32991.018
**Figure supplement 1.** Deletion of *Pitx2/3* in adult muscle satellite cells of *mdx* mice leads to aggravation of the dystrophic phenotype.
DOI: https://doi.org/10.7554/eLife.32991.017

*Figure 4—figure supplement 1A*). This is accompanied by dramatic muscle loss and premature death of double mutant mice (*Figure 4—figure supplement 1B*). Strikingly, treatment of *Pitx2^flox/flox^:Pitx3^flox/-^:mdx/mdx:Pax7^Cre-ERT2/+^* mice with NAC during the whole period (*Figure 4C*) prevented senescence (*Figure 4D*) and this marked aggravation of the dystrophic phenotype, leading to a

pattern of regeneration and a life expectancy similar to that of controls (*Figure 4—figure supplement 1*).

For tissue culture analysis of *Pitx2/3* double mutant satellite cells, Pax3(GFP)-positive cells were purified by flow cytometry from muscles of control *Pitx2$^{flox/+}$:Pitx3$^{+/-}$:Pax3$^{GFP/+}$:R26R$^{Cre-ERT2/+}$* and double mutant *Pitx2$^{flox/flox}$:Pitx3$^{flox/-}$:R26R$^{Cre-ERT2/+}$:Pax3$^{GFP/+}$* mice after five intra-peritoneal injections of 4-OHT (*Figure 5A*). Consistent with what we observed during foetal myogenesis (*L'honoré et al., 2014*), mutation of both *Pitx2* and *Pitx3* in satellite cells leads to altered expression of genes for antioxidant proteins (*Mgst2*, *SelX*, *SelM*, *Gstz*, *Hspb2*) and for the regulator of mitochondrial function *NRF1* (*Gleyzer et al., 2005*) (*Figure 5—figure supplement 1A*). We then assayed proliferation and differentiation properties of control and double mutant satellite cells. Similarly to what we observed with the *Pitx3* mutant, Pitx2/3 deficient satellite cells exhibit an expansion defect (*Figure 5B,C*). NAC addition to the culture medium alleviates this phenotype (*Figure 5B,C*), and also exerts a positive effect on control cells by improving both their cloning efficiency, probably reflecting improved survival (*Figure 5—figure supplement 1B*), and their expansion (*Figure 5B,C*). In contrast to Pitx3-deficient satellite cells, which undergo differentiation following premature cell cycle exit, mutation of both *Pitx2* and *Pitx3* genes leads to a marked reduction in differentiation markers (*Figure 5D,E*(i), *Figure 5—figure supplement 1G*), accumulation of carbonylated proteins (*Figure 5—figure supplement 1C,D*) and DNA damage marked by γH$_2$AX (*Bonner et al., 2008*) (*Figure 5E*(ii), *Figure 5—figure supplement 1G*). In contrast to the foetal situation (*L'honoré et al., 2014*), double mutant cells do not undergo apoptosis (data not shown) but become senescent, as evidenced by up-regulation of senescence associated genes (*Sousa-Victor et al., 2014*) (*Igfbp5*, *Ifitm1* or *Ccl5*, *Figure 5—figure supplement 1E*), and by the increased number of cells positive for the marker of heterochromatin foci Hp1γ (*Zhang et al., 2007*) (*Figure 5E*(iii), *Figure 5—figure supplement 1G*) and by senescence associated β-galactosidase staining (*Dimri et al., 1995*) (SA-βgal, *Figure 5—figure supplement 1F*), compared to controls. Culture of double mutant cells under 3% O$_2$ does not rescue the mutant phenotype. Treatment of control satellite cells purified from *Pax3$^{GFP/+}$* adult mice with buthionine sulfoximine (BSO) (*L'honoré et al., 2014*) leads to an increase in ROS levels and senescence (*Figure 5—figure supplement 1H,I*), suggesting that, while excessive ROS lead to apoptosis in foetal muscle cells (*L'honoré et al., 2014*), it triggers senescence in adult satellite cells. Treatment of double mutant cells with NAC efficiently prevents DNA damage (*Figure 5E*(ii), *Figure 5—figure supplement 1G*) and protein carbonylation (*Figure 5—figure supplement 1C,D*), resulting in markedly fewer senescent cells (*Figure 5E*(iii), *Figure 5—figure supplement 1F,G*), and in the restoration of their differentiation potential (*Figure 5E*(i), *Figure 5—figure supplement 1G*). In view of the role of p38α MAP kinase in senescence (*Coulthard et al., 2009*) and in single *Pitx3* mutant satellite cells (*Figure 3D–G*), we investigated the consequences of its inhibition in double mutant cells. Treatment with SB203580 efficiently reduces senescence (*Figure 5E*(iii), *Figure 5—figure supplement 1G*), but does not rescue DNA damage (*Figure 5E*(ii), *Figure 5—figure supplement 1G*) with consequent impairment of differentiation (*Figure 5E*(i), *Figure 5—figure supplement 1G*) (*Puri et al., 2002*). In the presence of both NAC and SB (NAC/SB), the effect is similar to that of NAC alone (*Figure 5E*, *Figure 5—figure supplement 1G*). Altogether, these data demonstrate that the excessive ROS levels observed in *Pitx2/3* double mutant satellite cells are responsible for their senescence, due to protein oxidation and DNA damage, a behaviour that probably accounts for the regeneration deficit observed in *Pitx2/3* mutant mice in vivo after injury (*Figure 4B*) or in dystrophic conditions (*Figure 4—figure supplement 1*).

## Manipulation of satellite cell redox state has implications for muscle cell therapy

In view of our results showing the impact of ROS levels on proliferation, differentiation and survival, we reasoned that manipulation of ROS levels and/or signalling by exogenous treatments could be an efficient way to expand wild-type satellite cells in culture, while preventing them from undergoing differentiation, two crucial features for maintaining their regenerative potential. Pax3(GFP)-positive satellite cells were cultured in the presence or absence of NAC, SB203580 (SB), or both compounds (NAC/SB). Immunostaining experiments (*Figure 6A*) show that while NAC or SB treatment alone leads to a reduction in p38α MAP kinase phosphorylation, the NAC/SB combination has a more pronounced effect on p38α MAP kinase phosphorylation and therefore activation of this signalling pathway. Accordingly, while NAC and SB treatments, under which proliferation is promoted (*Figure 6B*,

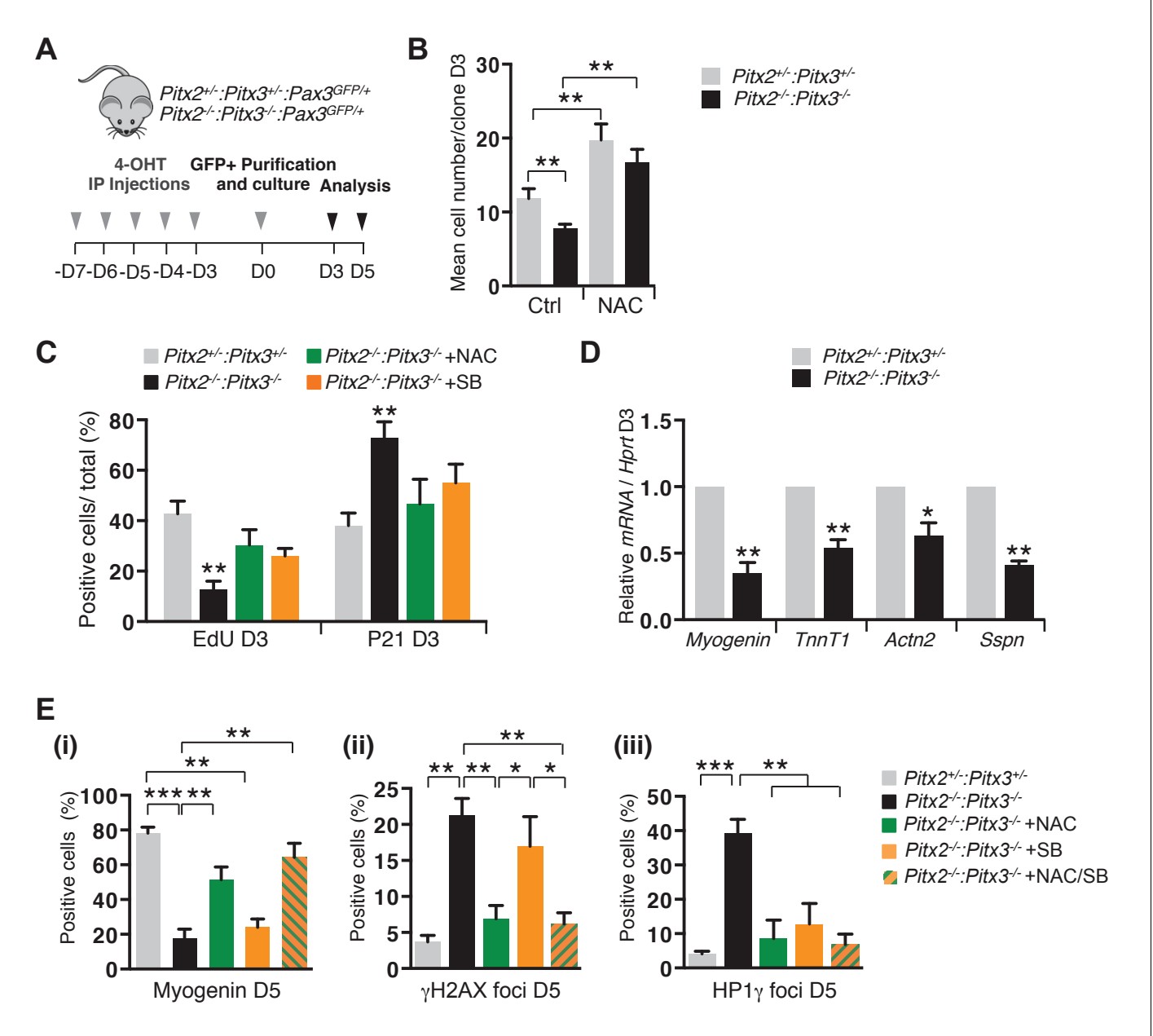

**Figure 5.** Excessive ROS levels cause premature cell cycle exit, differentiation defects and senescence of *Pitx2/3* double mutant satellite cells. (**A**) Double mutant *Pitx2^{flox/flox}:Pitx3^{flox/-}:Pax3^{GFP/+}:R26R^{Cre-ERT2/+}*, and control *Pitx2^{flox/+}:Pitx3^{+/-}:Pax3^{GFP/+}:R26R^{Cre-ERT2/+}* adult mice were obtained by 4-hydroxytamoxifen (4-OHT) intra-peritoneal (IP) injections for 5 consecutive days (D). Three days after the last injection, GFP-positive cells were isolated by flow cytometry from the *pectoralis*, abdominal and diaphragm muscles and analysed in culture. (**B**) control and mutant satellite cells were placed in culture and used for clonal analysis. Their proliferation capacity is expressed as the mean cell number per clone after 3 days of culture in the presence or absence of 10 mM N-Acetyl-Cysteine (NAC). (**C**) After 3 days of culture in the presence or absence of 10 mM NAC or 5 μM SB203580, cells were incubated with EdU and processed for detection of EdU incorporation and p21 expression. p21- and EdU-positive cells were counted, and quantified as the percentage (%) of total cells. (**D**) After 3 days of culture, total RNA samples were extracted from control and double mutant cells and analysed by qPCR for the expression of myogenic differentiation markers. Transcripts are shown relative to the level of *Hprt* transcripts, expressed as a log ratio. *TnnT1, Troponin T1; Actn2, Actinin α2; Sspn, Sarcospan.* (**E**) After 5 days of culture (D5) in the presence or absence of 10 mM NAC, 5 μM SB203580 or both NAC and SB203580, control and double mutant cells, marked by DAPI staining, were analysed by immunofluorescence with Myogenin, HP1γ and γH2AX antibodies. Myogenin-positive cells (**i**), γH2AX-foci (**ii**) and HP1γ foci-positive cells (**iii**), were counted, and quantified as the percentage (%) of total cells. (**B–E**) Error bars represent the mean ± s.d, with n = 3 animals for each genotype and each condition, *p<0.05, **p<0.01, ***p<0.001. Please see *Figure 5—figure supplement 1* for additional data.

DOI: https://doi.org/10.7554/eLife.32991.019

*Figure 5 continued on next page*

*Figure 5 continued*

The following source data and figure supplements are available for figure 5:

**Source data 1.** Numerical data used to generate *Figure 5*.

DOI: https://doi.org/10.7554/eLife.32991.022

**Figure supplement 1.** Deletion of *Pitx2/3* in adult muscle satellite cells leads to altered expression of genes encoding antioxidant enzymes and accumulation of DNA damage.

DOI: https://doi.org/10.7554/eLife.32991.020

**Figure supplement 1—source data 1.** Numerical data used to generate *Figure 5—figure supplement 1*.

DOI: https://doi.org/10.7554/eLife.32991.021

*C*), both reduce differentiation (*Figure 6D*), treatment with the dual NAC/SB combination leads to even stronger effects (*Figure 6D*). Indeed, whereas after 3 days of culture in control conditions only 40% of cells show EdU incorporation, treatment with the NAC/SB combination leads to a major increase, with 80% of cells positive for EdU (*Figure 6E*, *Figure 6—figure supplement 1A*(i)). Clonal analysis shows that this strong proliferative activity promotes a 10 fold increased expansion compared to controls, with a mean number of 250 cells per clone compared to 25 (*Figure 6B*). After 3 days of culture, NAC/SB-treated cells do not show p21 and Myogenin expression (*Figure 6E*, *Figure 6—figure supplement 1A*), and retain a high level of Pax7 expression (data not shown), consistent with the absence of differentiation. Similar results were obtained when cells were cultured in 3% $O_2$ (*Figure 6—figure supplement 1B,C*), showing that conditions considered as normoxic do not substitute for NAC.

Neither mitochondrial mass (*Figure 6—figure supplement 1D*) nor MyoD expression (*Figure 6—figure supplement 1A*(ii)) were significantly affected under these conditions. It is notable that SB has little effect on ROS levels compared to NAC (*Figure 6—figure supplement 1E*), which is also reflected in cloning efficiency where NAC promotes cell viability and prevents senescence by reducing ROS levels (*Figure 6—figure supplement 1F*). In view of the synergistic effect seen with NAC and SB together (*Figure 6E*), we checked whether ERK, JUNK, p65 (NFκB) and ATM were affected and found no evidence that these signalling pathways were implicated (*Figure 6—figure supplement 1G,H*, data not shown).

In view of these results, we anticipated that NAC/SB treatment, in addition to promoting expansion of satellite cells in culture, might improve their regenerative potential in vivo, by maintaining them in a highly proliferative and undifferentiated state. To test this hypothesis, 15.000 freshly isolated Pax3(GFP)-positive satellite cells, or 15.000 Pax3(GFP)-positive satellite cells cultured for three days (D3) in the absence or in the presence of NAC/SB, were grafted into the *Tibialis Anterior* muscle of $Rag2^{-/-}$:$Il2r\beta^{-/-}$:$dmd/dmd$ mice (*Vallese et al., 2013*) (*Figure 6—figure supplement 1I*). Three weeks after grafting, muscle sections were assayed for Dystrophin expression by immunofluorescence (*Figure 6F*). While satellite cells cultured under standard conditions display a regenerative potential at least 10 times lower than freshly isolated satellite cells (*Montarras et al., 2005*) (*Figure 6F,G*), NAC/SB treatment of cultured satellite cells leads to a very efficient engraftment, with a number of Dystrophin-positive cells similar to that obtained with freshly isolated satellite cells (*Figure 6F,G*). To test whether engrafted cells also contributed to the satellite cell compartment in recipient muscles, we performed immunostaining with GFP and Pax7 antibodies. Both staining colocalized in a subset of nuclei in the engrafted muscle, indicating that the transplanted cells also successfully populated the satellite cell compartment (*Figure 6—figure supplement 1J*). To address the issue of the expansion and self-renewal of control and treated cells, 15.000 cells cultured for three days in control, NAC, SB or NAC/SB conditions were grafted into the *Tibialis Anterior* muscle of $Rag2^{-/-}$:$Il2r\beta^{-/-}$:$dmd/dmd$ mice. Injury regeneration assays performed 3 weeks after grafting (*Figure 6H–K*) show that donor satellite cells treated with both NAC and SB203580 are much more abundant, (x20, *Figure 6K*) than when treated with NAC or SB alone, which also have a positive effect compared to controls (x2, *Figure 6I*). When put in culture, all the Pax3(GFP)-positive cells recovered from grafted muscles displayed a MyoD-positive phenotype (data not shown), confirming their myogenic identity. In addition, this grafting analysis (*Figure 6J,K*) further indicates that while NAC or SB treatments both improve the regeneration potential of cultured satellite cells (*Figure 6J, K*), the use of both compounds leads to a much higher number of Dystrophin-positive fibres (*Figure 6J,K*). Taken together, these results indicate that cells treated with both compounds display

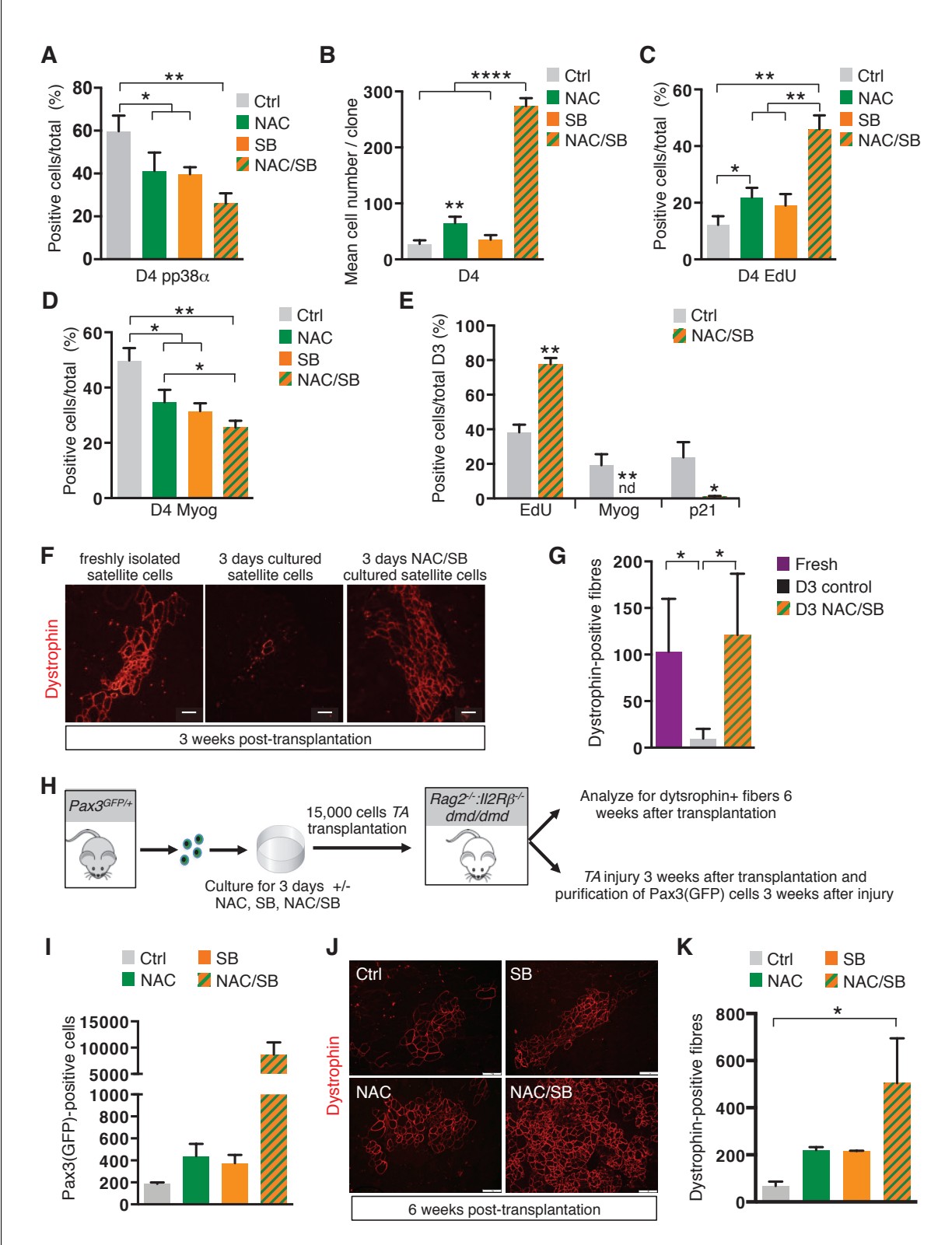

**Figure 6.** The transition of satellite cells from proliferation to differentiation is regulated by their redox state. (A–E) GFP-positive cells isolated by flow cytometry from the *pectoralis*, abdominal and diaphragm muscles of *Pax3*$^{GFP/+}$ mice were cultured for 4 days (D4) in the absence (ctrl) or in the presence of 10 mM N-Acetyl Cysteine (NAC), or 5 µM SB203580 (SB), or of both NAC and SB203580 (NAC/SB) in mass (A, C, D) or clonal conditions (B). (A) After 4 days in culture, cells were processed for phospho-p38α (pp38α) expression by immunofluorescence. pp38α-positive cells were counted, and

*Figure 6 continued on next page*

*Figure 6 continued*
quantified as the percentage (%) of total cells. (B) The cell proliferation capacity is expressed as the mean cell number per clone. (C, D) After 4 days in culture, cells were incubated with EdU and processed for EdU detection by fluorescence (C) or for Myogenin (Myog) detection by immunofluorescence (D). EdU-positive cells (C) and Myogenin-positive cells (D) were counted and quantified as the percentage (%) of total cells. (E) GFP-positive cells isolated as in (A) were cultured for 3 days (D3) in the absence (ctrl) or in the presence of both NAC and SB203580 (NAC/SB). Cells were incubated with EdU and processed for EdU detection by fluorescence, and for Myogenin or P21 detection by immunofluorescence. Myogenin, EdU and p21 positive cells were counted, and quantified as the percentage (%) of total cells. (A–E) The experiments were performed with n $\geq$ 3 animals for each condition. Error bars represent the mean +/- s.d, with *p<0.05, **p<0.01, ****p<0.001. (F–G) *Tibialis Anterior* (TA) muscles of *Rag2$^{-/-}$:Il2rβ$^{-/-}$:dmd/dmd* mice were irradiated and grafted with 15.000 cells freshly isolated from *Pax3$^{GFP/+}$* mice (Fresh) or cultured for 3 days in the absence (D3) or in the presence of both NAC and SB203580 (D3 NAC/SB). Three weeks after grafting, TA muscles were dissected and cryo-sections analysed by immunofluorescence. (F) Immunofluorescence for Dystrophin and (G) the number of Dystrophin-positive fibres was counted for each condition. (F–G) The experiments were performed with n $\geq$ 3 grafted TAs for each condition, and a representative image is shown, scale bar, 200 μm (F). (H–K) *Tibialis Anterior* (TA) muscles of *Rag2$^{-/-}$:IlrRβ$^{-/-}$:dmd/dmd* mice were irradiated and grafted with 15.000 cells isolated from *Pax3$^{GFP/+}$* mice and cultured for 3 days in the absence (ctrl) or in the presence of NAC, SB203580 (SB), or both NAC and SB203580 (NAC/SB). Three weeks after grafting, for each mouse, one TA muscle was subjected to injury by cardiotoxin injection, and the contralateral muscle was used as a control. Six weeks after grafting, all TA muscles were dissected. (I) Injured TA muscles were used for Pax3(GFP)-positive cell purification by flow cytometry and the number of cells counted for each condition. Non-injured TA muscles were analysed by immunofluorescence on cryo-sections (J) and the number of Dystrophin-positive fibres was counted for each condition (K). (H–K) The experiments were performed with n $\geq$ 2 grafted TAs for each condition, and a representative image is shown, scale bar, 200 μm (J). Error bars in (G, I, K) represent the mean number of Dystrophin-positive fibres per section (G, K) or of Pax3(GFP)-positive cells per muscle (I) $_+$ s.d, with *p<0.05. Please see *Figure 6—figure supplement 1* for additional data.

DOI: https://doi.org/10.7554/eLife.32991.023

The following source data and figure supplements are available for figure 6:

**Source data 1.** Numerical data used to generate *Figure 6*.
DOI: https://doi.org/10.7554/eLife.32991.026

**Figure supplement 1.** Alteration of the redox state of satellite cells modulates their survival and amplification in culture and their regenerative potential after grafting.
DOI: https://doi.org/10.7554/eLife.32991.024

**Figure supplement 1—source data 1.** Numerical data used to generate *Figure 6—figure supplement 1*.
DOI: https://doi.org/10.7554/eLife.32991.025

a higher regeneration potential, together with a much higher capacity to expand and self-renew upon grafting.

## Discussion

Our analysis of *Pitx2* and *Pitx3* mutant mice demonstrates the importance of these transcription factors in regulating the behaviour of adult muscle stem cells and consequently the regeneration of skeletal muscle. In the double mutant, these satellite cells undergo DNA damage and become senescent. This is in contrast to the situation in the embryo where muscle progenitor cells respond to DNA damage by undergoing apoptosis in the absence of both factors (*L'honoré et al., 2014*). As during foetal myogenesis, we show that Pitx2/3 factors are involved in the redox control of the muscle satellite cell. The link between Pitx factors and the control of ROS has also been reported in cardiomyocytes where *Pitx2*, the only *Pitx* gene expressed in the heart, plays a critical role in the regulation of antioxidant scavenger genes (*Tao et al., 2016*).

In the absence of Pitx3 alone, the balance between proliferation and differentiation is perturbed so that satellite cells differentiate prematurely, a phenotype that impacts the efficiency of muscle regeneration. This mutant provides a valuable tool to investigate the regulation of this critical balance. Although historically viewed as harmful, ROS are now established as important regulators of signalling pathways (*Bigarella et al., 2014*; *Holmström and Finkel, 2014*; *Khacho and Slack, 2017*). Our analysis of skeletal muscle regeneration in the *Pitx3* mutant reveals a critical role for the redox state of satellite cells in the regulation of their behaviour (*Figure 7*). While quiescent satellite cells exhibit low ROS levels, we observed a marked increase in activated cells at the onset of differentiation. Using chemical as well as genetic manipulation of the redox state of satellite cells during muscle regeneration, we show that the increase in ROS occurring in proliferating satellite cells leads to activation of p38α MAP kinase and link this redox activation to its function as a critical regulator of satellite cell differentiation (*Segalés et al., 2016*; *Perdiguero et al., 2007*; *Brien et al., 2013*;

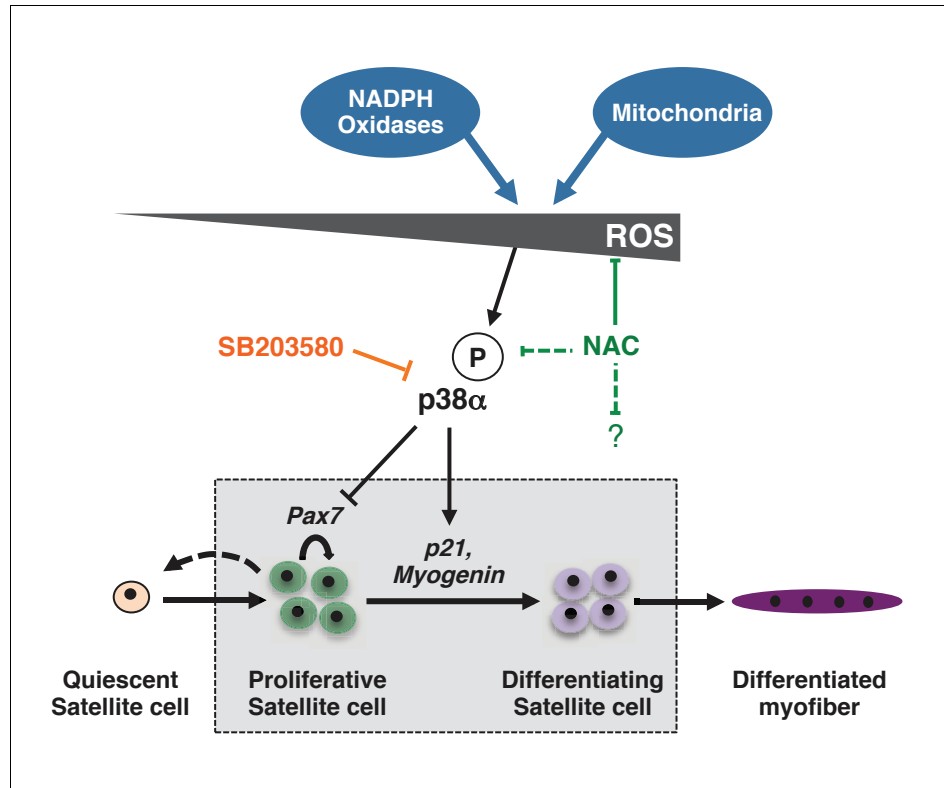

**Figure 7.** Model for the regulation of proliferation and differentiation of muscle stem cells by their redox state. This model represents redox regulation acting through p38α MAP kinase of a wild-type satellite cell as it progresses towards a differentiated myofibre. ROS levels, as indicated by effects observed for NAC but not SB, also probably impact the satellite cell, independently of p38α signalling. Please see *Figure 7—figure supplement 1* for additional data.

DOI: https://doi.org/10.7554/eLife.32991.027

The following figure supplement is available for figure 7:

**Figure supplement 1.** Consequences of altered ROS levels on satellite cell function during skeletal muscle regeneration.

DOI: https://doi.org/10.7554/eLife.32991.028

*Palacios et al., 2010*) (*Figure 7*, *Figure 7—figure supplement 1*). Given the striking synergistic effect that we observe with NAC which lowers ROS levels and SB which inhibits p38α MAP kinase, it is probable that ROS levels also affect satellite cell behaviour independently of p38α MAP kinase. This is demonstrated by our analysis of the double *Pitx2/3* mutant, where DNA damage is reversed by NAC treatment but not by SB, indicating that they reflect a p38α−independent effect of high ROS levels. In wild-type satellite cells too, cloning efficiency is improved by NAC but not SB treatment suggesting that lowering ROS levels in normal cells also reduces a background level of DNA damage. The striking effect of NAC and SB together may reflect reduced DNA damage due to ROS, which will repercuss on the cell cycle, as well as delayed differentiation (*Shaltiel et al., 2015*).

## Redox regulation of muscle stem cells during muscle regeneration

ROS are mainly generated by the mitochondrial respiratory chain and by NADPH oxidases (*Murphy, 2009*; *Katsuyama et al., 2012*). While quiescent satellite cells, with low energetic requirements, are characterized by reduced mitochondrial mass and low ROS production, their activated progeny is marked by increased mitochondrial mass and activity and enhanced ROS levels. We show that a peak of mitochondrial ROS production occurs at day 2 of regeneration, when mitochondrial mass is only beginning to increase in activated satellite cells. Under conditions in which the cellular energy demand is still moderate, increased ROS production has been reported to occur from complex I (*Murphy, 2009*). In addition to mitochondrial ROS, by using Apocynin, an inhibitor of

NADPH oxidases (*Petrônio et al., 2013*), we demonstrate a contribution of these enzymes to the redox state of satellite cells and the regulation of satellite cell differentiation. In this study we have focused on the cell-autonomous role of the satellite cell redox state, but extracellular sources of ROS released into the environment by injured muscle fibres and inflammatory cells (*Le Moal et al., 2017*), will also contribute to the in vivo scenario.

## Interplay between redox regulation, metabolic reprogramming and epigenetic modifications

The metabolic switch that accompanies the satellite cell transition from quiescence to ex vivo activation (*Ryall et al., 2015*) results in a decrease of intracellular NAD(+) levels and of the activity of the histone deacetylase, Sirt1, leading to elevated H4K16 acetylation and activation of muscle genes required for the entry of satellite cells into the myogenic program (*Ryall et al., 2015*). In the hematopoietic system, Sirt1 has been shown to be critical for the maintenance of hematopoietic stem cells (*Matsui et al., 2012*). The premature differentiation seen in *Sirt1* mutant satellite cells (*Ryall et al., 2015*), resembles what we observe in the *Pitx3* mutant (*Figure 7—figure supplement 1*). In addition, Sirt1 has been characterized as a direct redox sensor, which is inactivated upon oxidation of its cysteine residues (*Zee et al., 2010*). It is therefore possible that increased ROS levels during normal regeneration lead to Sirt1 inactivation in parallel with activation of p38α MAP kinase, and consequent onset of differentiation. This parallel action may contribute to the more striking effect that we observe with NAC, which reduces ROS levels with a consequent increase in Sirt1 activity, compared to the effect of inhibiting p38α MAP kinase with SB203580 alone, which does not correct the DNA damage and only partially corrects the differentiation defect seen in *Pitx2/3* double mutant mice. It should be noted that the level of SB203580 that we employ (5 µM) is lower than that used by other authors (10 to 25 µM) who have reported beneficial effects of this drug on the regenerative properties of satellite cells (*Bernet et al., 2014*; *Charville et al., 2015*). We chose to use a lower dose because of off-targets effects of the drug on other signalling pathways (*Bain et al., 2007*). The effects that we observe on p38α phosphorylation in wild type cells are similar with SB203580 and with NAC, indicating that the dose of SB203580 is effective. SB203580 acts on threonine 106 to inhibit phosphorylation, whereas ROS, and therefore the ROS scavenger NAC, can affect the redox state of cysteine residues involved in substrate binding and activation of p38α MAP kinase (*Bassi et al., 2017*). Additional effects of NAC due to the lowering of ROS levels, through a reduction in DNA damage and a potential increase in Sirt1 activity, are independent of p38α MAPK kinase (*Figure 7*).

## Muscle stem cell ageing and perturbation of redox homeostasis

In skeletal muscle, as in many tissues, ageing is accompanied by a decline in regenerative capacity (*Sousa-Victor et al., 2014*; *Cosgrove et al., 2014*). While the aged systemic environment significantly impacts the function of satellite cells (*Gopinath and Rando, 2008*), recent studies have shown that impaired regeneration is also due to a cell-autonomous functional decline in satellite cells (*Sousa-Victor et al., 2014*; *Cosgrove et al., 2014*; *Bernet et al., 2014*). Skeletal muscle is a remarkably stable tissue, and satellite cells are long-lived resident adult stem cells. Despite their low metabolic rate and their high antioxidant capacity (*Pallafacchina et al., 2010*), these long-lived cells are particularly exposed to ROS generated either by mitochondria during muscle fibre contraction (*Jackson, 2011*), or by NADPH oxidases shown to be over-expressed in sarcopenic fibres (*Kerkweg et al., 2007*). This exposure to ROS throughout life combined with a decline in their antioxidant capacity during ageing (*Sullivan-Gunn and Lewandowski, 2013*; *Fulle et al., 2005*) makes geriatric satellite cells particularly vulnerable to accumulation of genomic and mitochondrial DNA mutations, and as a consequence, to mitochondrial dysfunction and genomic instability. Geriatric satellite cells have been shown to have elevated p38α MAP kinase activity (*Cosgrove et al., 2014*; *Bernet et al., 2014*) and defective autophagy leading to mitochondrial dysfunction, excessive ROS levels (*García-Prat et al., 2016*), with increased DNA damage (*García-Prat et al., 2016*; *Sinha et al., 2014*) and senescence (*Sousa-Victor et al., 2014*; *Latella et al., 2017*). In view of the *Pitx2/3*-mutant phenotype (*Figure 7—figure supplement 1*), and in correlation with observations on hyper-homocysteinemic mice (*Veeranki et al., 2015*), we propose that *Pitx2/3* mutants may represent a useful model of premature muscle ageing. Future experiments will address this issue.

## Redox homeostasis and stem cell therapies

A major obstacle to the development of stem cell based therapies is the number of cells required for transplantation (*Briggs and Morgan, 2013*; *Walasek et al., 2012*). Ex vivo amplification of tissue specific stem cells is an option, but it generally leads to a reduction of their regenerative potential, due to their commitment to differentiation (*Montarras et al., 2005*; *Walasek et al., 2012*; *Briggs and Morgan, 2013*). In the muscle field, advances have been made by manipulating dystrophic muscle (*Price et al., 2014*; *Tierney et al., 2014*), by modifying culture conditions (*Gilbert et al., 2010*; *Liu et al., 2012*; *Parker et al., 2012*; *Charville et al., 2015*), or by grafting selected satellite cell sub-populations (*Jean et al., 2011*), however further improvements are mandatory. Treatment with SB203580, that inhibits p38α MAP kinase activation required for the onset of differentiation, promotes proliferation of human myoblasts in culture and their contribution to the reconstitution of the satellite cell pool upon grafting into injured skeletal muscles (*Charville et al., 2015*). NAC treatment of Pitx2 deficient mice restores the regenerative capacity of the newborn heart (*Tao et al., 2016*), a situation that is reminiscent of our finding for skeletal muscle in triple Pitx2/3:Dystrophin-deficient mice, where the dystrophic muscle phenotype is improved by NAC treatment. We now show that the use of the NAC/SB203580 combination in culture not only leads to fast and very efficient amplification of satellite cells, but also maintains their regenerative potential at levels similar to that of freshly isolated cells in grafting experiments. In vivo also, treatment with both compounds promotes expansion of the transplanted satellite cell pool after injury. These observations have important implications for stem cell therapy.

## Materials and methods

### Animal breeding, Cre recombination and genotyping

All animal procedures were approved and conducted in accordance with the Institut Pasteur animal ethics committee (CEEA Institut Pasteur n°2013–0017 and APAFIS #2455 2015 1122133311) following the regulations of the Ministry of Agriculture and the European Community guidelines.

*Pitx2*-floxed (*Pitx2^{flox/+}*) mice were provided by S. Camper and P. Gage. *Pitx3-floxed(neo)* (*Pitx3-^{flox}*), *Pitx3*-null (Pitx3⁻) and Pax3-GFP (*Pax3^{GFP/+}*) alleles have been respectively described (*L'honoré et al., 2014*; *Montarras et al., 2005*). *Pax7^{Cre-ERT2/+}* mice were provided by C.M.Fan and C.Lepper. *R26R^{CRE-ERT2/+}* and *mdx/mdx* mice were obtained from the Jackson laboratories. *Rag2⁻/⁻:Il2rβ^{-/-}:dmd/dmd* mice were provided by V.Mouly and G.Butler-Browne (*Vallese et al., 2013*). Genotyping was carried out by PCR using genomic DNA isolated from adult tail sections, with previously described protocols and primers: *L'Honoré et al. (2007)* for *Pitx3^{flox}* and *Pitx3⁻* alleles; *L'honoré et al. (2014)* for *Pitx2^{flox}*, *Pitx2⁻* and *R26R^{Cre-ERT2}* alleles; *Coulthard et al. (2009)* for *Pax7-^{Cre-ERT2}* allele; *Relaix et al. (2006)* for *Pax3^{GFP}* allele, *Bulfield et al. (1984)* for the *mdx* allele and *Vallese et al. (2013)* for *Rag2⁻*, *Il2rb⁻*, and *dmd* alleles.

Recombination of floxed alleles was obtained by five daily intra-peritoneal injections of 4-hydroxy-tamoxifen (4-OHT, Sigma). 4-OHT was prepared by dissolving a freshly opened bottle of 50 mg in EtOH 100% at 80 mg/ml. The mixture was incubated at 55°C with constant agitation until 4-OHT was completely dissolved (~1–2 hr). After complete solubilisation, 4-OHT was diluted 1:2 with Cremophor EL (Sigma) and kept at 4°C. Just before injection, 4-OHT from this stock was diluted 1:4 in PBS and administered to 6 to 8 weeks-old control and mutant mice (1 mg final 4-OHT/mouse of 25 g). Recombination efficiency was verified by PCR using genomic DNA isolated from purified satellite cells three days after the last injection.

Description of mouse lines and their use are given in *Supplementary file 1*. Briefly, experiments that do not require satellite cell purification were performed on animals in which *Pitx2/3* genes were conditionally recombined using the *Pax7^{Cre-ERT2/+}* mice line, except for experiments using the single constitutive mutant *Pitx3*. As *Pax3^{GFP/+}:Pax7^{Cre-ERT2/+}* animals are not viable due to non-muscle defects, we used the inducible *R26R^{CRE-ERT2/+}* mouse line for all experiments requiring control and mutant satellite cell purification by flow cytometry, except for experiments using the single constitutive mutant *Pitx3*.

Unless indicated, all experiments were performed on 6 to 8 weeks-old adult animals.

N-Acetyl-Cysteine (NAC, Sigma) was diluted just prior to use at 500 mM in $H_2O$ in light-protected bottles (*Richards et al., 2011*) and given ad libitum to mice, with the solution being changed every three days.

## Muscle injury

Mice were anesthetised with 0.5% Ketamine (Imalgene1000, 100 mg/kg, Merial) and 0.5% Xylazine (Rompun 2%, 20 mg/kg, Bayer). *Pectoralis* and *Tibialis Anterior* muscles were injected with 10 µl of notexin (12.5 µg/ml, Latoxan) and 20 µl of cardiotoxin (20 µg/ml, Latoxan) respectively. After injection, mice were kept on a warm plate until recovery. Mice were sacrificed by cervical dislocation at different days after injury and the muscles used for analysis by histology, immunofluorescence or for satellite cell purification.

Regeneration experiments followed by histology and immunofluorescence were performed on the *Tibialis Anterior* (*TA*) muscle injected with cardiotoxin. As Pax3(GFP) expressing satellite cells are much less abundant in *TA* muscle (*Relaix et al., 2006*), purification of activated satellite cells during regeneration was performed at different days after intramuscular injection of the *pectoralis* muscle with notexin. Notexin was used in this case in order to obtain a more widespread injury than with cardiotoxin (*Hardy et al., 2016*).

## Primary cell purification and culture

Satellite cells were prepared and cultured as previously described (*Montarras et al., 2005*) with minor modifications. For primary cell culture, dissected diaphragm, abdominal and *pectoralis* muscles of independent adults (6–8 weeks old mice) were incubated in digestion buffer (0.1% Trypsin, 0.1% Collagenase V [Roche]) at 37°C with frequent agitation. After successive digestions, pooled supernatants were centrifuged at 1800 rpm for 20 min and cells were purified by flow cytometry via MoFlow Astrios (Beckman Coulter) on the basis of size, granularity and GFP fluorescence. A mean of 70.000 to 100.000 Pax3(GFP)-positive cells were obtained per adult mouse. 5.000 Pax3(GFP)-positive cells were seeded on 35 mm plastic dishes pre-coated with gelatin. Cells were cultured in growth medium (40% F12, 40% DMEM, 20% SVF, 2% Ultroser [Biosepra]) and left to differentiate spontaneously. When cultured under 3% $O_2$, cells were seeded immediately after purification in a Hypoxia Chamber Workstation (SCI-tive, Baker Ruskinn) at 37°C, 5% $CO_2$ and 70% humidity. Each experiment corresponded to a minimum of three individual mice that were treated and analysed separately. Clonal analysis was performed as previously described (*Pallafacchina et al., 2010*). For each genotype, two 96 plates were analysed for each condition (in the absence or presence of N-Acetyl-Cysteine, SB20380 or both) for three individual mice.

For purification of satellite cells during regeneration, a similar protocol was used with minor modifications: dissected *pectoralis* muscles from different adult mice were incubated in digestion buffer (0.1% Trypsin, 0.1% Collagenase V [Roche]) at 37°C with low agitation. After successive digestions, pooled supernatants were centrifuged at 1800 rpm for 20 min and cells were purified by flow cytometry via MoFlow Astrios (Beckman Coulter) on the basis of size, granularity and GFP fluorescence. A mean of 10.000 to 30.000 Pax3(GFP)-positive cells were obtained per muscle. When the number of purified Pax3(GFP)-positive cells was too low to get significant results, several animals were pooled in order to obtain each biological sample.

N-Acetyl-Cysteine (Sigma), Trolox (Abcam) and SB203580 (InVivoGen) were used for cell seeding at 10 mM, 10 µM, and 5 µM respectively in growth media. Apocynin (Santa Cruz Biotechnology), Buthionine Sulfoximine (Enzo Life Sciences) and TNFα (R and D systems) were added to the cell culture medium at 10 mM, 10 µM and 0.5 nM respectively.

## Satellite cell transplantation

Three months-old immune-compromised $Rag2^{-/-}:Il2r\beta^{-/-}:dmd/dmd$ mice (*Vallese et al., 2013*) were subjected to leg-X ray irradiation using a Faxitron equipment 4 days before cell engraftment. For each injection, 15.000 satellite cells from $Pax3^{GFP/+}$ mice, collected by flow cytometry (freshly isolated cells) or obtained by trypsination after 3 days of culture (cultured cells), were resuspended in 15 µl of PBS and injected into the *Tibialis Anterior* muscle with a 25 µl Hamilton syringe. Three or six weeks after transplantation, mice were sacrificed by cervical dislocation and the muscle analysed.

## Immunofluorescence

Immunofluorescence on frozen muscle sections and on primary cell cultures was carried out as previously described (*Pallafacchina et al., 2010*). The following primary antibodies were used for immunohistochemistry and immunofluorescence on frozen muscle sections and on primary cell cultures: polyclonal anti-Pitx2 1:200 and polyclonal anti-Pitx3 1:300 (*L'Honoré et al., 2007*), monoclonal anti-p21 1:100 (Pharmingen), rabbit anti-Pax7 1:1000 (Aviva ARP32742), monoclonal anti-MyoD 1:100 (Pharmingen), monoclonal anti-Myogenin 1:100 (Pharmingen), polyclonal anti-Myogenin 1:100 (SantaCruz), polyclonal anti-MyoD 1:100 (Santacruz) monoclonal anti-embryonic MyHC 1:20 (DSHB), monoclonal anti-Troponin T 1:300 (T6277, Sigma), monoclonal anti-$\gamma$H2AX 1:300 (JBW301, Millipore), monoclonal anti-HP1$\gamma$ 1:3000 (Sigma) and monoclonal anti-dystrophin 1:50 (Sigma). Secondary antibody detection was carried out with Alexa-conjugated antibodies 1:150 (Alexa 488 or Alexa 546).

For EdU staining, the Click-iT EdU Cell Assay kit (Invitrogen) was used following the manufacturer's instructions with minor modifications: EdU was incubated at 10 $\mu$M for 15 min in cell culture medium. For detection, immunostaining for primary (Myogenin on cell culture and Pax7 on muscle sections) and secondary antibodies was performed, followed by the click chemical reaction using Alexa488 as a reactive fluorophore for detection, followed by DAPI staining.

Senescence associated $\beta-$galactosidase assays were performed as previously described (*Dimri et al., 1995*) on cultured cells, and on frozen muscle sections.

## RNA extraction and RT-qPCR analysis

Cells were collected after culture or after flow cytometry in RLT buffer (Qiagen). Total RNA was extracted and purified after DNase treatment (Promega) using the RNAeasy Micro kit (Qiagen) and was reverse-transcribed into cDNA, following instructions in the Transcriptor cDNA synthesis kit (Roche). Quantitative real time PCR was performed on a StepOnePlus PCR machine (Applied Biosystems), using the FastStart Universal SYBR Green Master (Roche). Relative levels of expression in each assay were obtained through the $\Delta\Delta$Ct method, and are expressed on a log scale. *Hprt* and *RPLO* transcripts levels were used for the normalizations of each target. As similar results were obtained with both, normalization has been done with *Hprt*. At least three biological replicates were used for each condition. Custom primers were designed using the Idt online software. Sequences are provided in *Supplementary file 2*. For each primer set, dissociation experiments were performed to ensure that no primer dimers or false amplicons would interfere with the results, and amplification efficiency was calculated using serial dilutions of total cDNA.

## Quantification of mitochondrial DNA

Cells were collected after flow cytometry in extraction buffer (0.2 mg/ml proteinase K, 0.2% SDS and 5 mM EDTA) and incubated at 50°C for 3 hr. DNA was precipitated with 0.3 M sodium acetate (pH 5.2) and isopropanol for 20 min on ice before centrifugation at 12.000 rpm at 4°C. Quantitative real time PCR was performed on a StepOnePlus PCR machine (Applied Biosystems) using the FastStart Universal SYBR Green Master (Roche) and primers that amplify the mitochondrial *COI* gene and the nuclear encoded *NDUFV1* gene as an endogenous reference (see *Supplementary file 2*).

## Metabolic and bio-energetic analysis

$H_2O_2$ production was quantified by the Amplex red assay (Life technology), which detects the accumulation of a fluorescent oxidized product.

Cytoplasmic reactive oxygen species were quantified by flow cytometry using the Deep Red Cell Rox probe (Invitrogen). For analysis of cells in culture, Cell Rox probe (0.5 mM) was directly added to the culture medium at the indicated time for 30 min in the cell culture incubator (at 37°C and in the dark). After three rinses with PBS at 37°C, cells were then rapidly trypsinised, fixed for 5 min in PFA, and rinsed three times in PBS. Cells were finally re-suspended in PBS and immediately analysed on a LSR Fortessa (BD Biosciences) apparatus, using FlowJo (TreeStar) software. For analysis of cells after flow cytometry, freshly sorted cells were re-suspended in 2 ml of F12 medium with 1% FCS at 37°C and incubated for 30 min at 37°C in a cell culture incubator in the presence of the Deep Red Cell Rox probe (0.5 mM). Cells were then rinsed in PBS at 37°C, fixed for 5 min in PFA for the Cell

Rox probe, rinsed 3 times in PBS, and re-suspended in PBS for analysis on a LSR Fortessa (BD Biosciences).

Mitochondrial reactive oxygen species were quantified using Mitosox probe (Invitrogen): freshly sorted cells were re-suspended in 2 ml of HBSS medium with 1% FCS at 37°C and incubated for 20 min at 37°C in a cell culture incubator in the presence of the Mitosox probe (0.5 mM). Cells were then rinsed in HBSS at 37°C, and immediately analysed on a LSR Fortessa (BD Biosciences).

Mitochondrial mass was quantified using Mitotracker RedSox probe (Invitrogen): freshly sorted cells were re-suspended in 2 ml of F12 medium with 1% FCS at 37°C and incubated for 30 min at 37°C in a cell culture incubator in the presence of the Mitotracker RedSox probe (250 nM). Cells were then rinsed in PBS at 37°C, and immediately analysed on a LSR Fortessa (BD Biosciences).

Cellular bioenergetics was analysed on a Seahorse extracellular flux analyser (XFe24) according to the manufacturer's instructions. Briefly, Pax3(GFP)-positive cells purified by flow cytometry were seeded at a density of 100.000 cells per well in a Seahorse XF24 plate coated with the cell immobilizer BD Cell Tak coating (BD Biosciences) and cultured for 2 to 4 days. The seeding density was based on initial assays that optimized the oxygen consumption rate (OCR) and the extracellular-acidification rate (ECAR). One hour before the assay, cultured cells were washed three times in minimal assay media (Seahorse Biosciences, 37°C, pH 7.40) supplemented with glucose 2.25 g/L, 1 mM sodium pyruvate and 2 mM L-glutamine, and equilibrated for 1 hr at 37°C in a non-$CO_2$ incubator. The basal and maximal rate of OCR and ECAR were measured successively in the control condition (basal rates) and after simultaneous injection of oligomycin (1 μM) that inhibits ATP synthase and of FCCP (0.5 μM) that uncouples mitochondrial OXPHOS (maximal rates). This dual injection allows measurement of the ability of cells to respond to increased energy demand using both oxidative phosphorylation (OCR) and glycolysis (ECAR). At the end of the experiment, proteins were extracted using 10 μL of lysis buffer and quantified using the Bradford assay. OCR and ECAR values were then normalized to protein concentration in order to compare different conditions.

## Western-blots

The following primary antibodies were used for western-blot: rabbit polyclonal anti-p38a 1:500 (Santacruz), mouse monoclonal anti-phospho-p38α 1:1000 (T180/Y182, Cell Signaling, #9216S), mouse monoclonal anti-ERK 1:500 (Cell Signaling, L34F12), rabbit polyclonal anti-phospho-ERK 1:500 (T202/Y204, Cell Signaling, #9101S), rabbit polyclonal anti-JUNK 1:500 (Cell Signaling, #9252), mouse monoclonal anti-phospho-JUNK 1:500 (T183/Y185, Cell Signaling, G9), rabbit monoclonal anti-ATM (Cell Signaling, D2E2), mouse monoclonal anti-phospho-ATM (Cell Signaling, 10H11.E12), rabbit monoclonal anti-NF-κB p65 1:500 (Cell Signaling, D14E12), mouse monoclonal anti-phospho-NF-κB p65 1:500 (S536, Cell Signaling, 7F1) and polyclonal anti-GAPDH 1:2000 (G9545, Sigma).

## Oxyblots

Irreversible protein oxidations were quantified by determining levels of protein carbonylation using the Protein OxyBlot kit (Millipore). Briefly, after protein extraction using lysis buffer, 10 μg of total proteins (in 5 μL) were denatured by adding 5 μL of 12% SDS and boiling at 95°C for 15 min. Derivatization was achieved by the addition of 5 μL DNPH (10 mM) and incubation at room temperature for 20 min. After neutralization by addition of 7.5 μL of neutralization solution, samples were loaded on 4–10% polyacrylamide gels. For controls, the same samples were treated as described except that a control solution without DNPH was used instead of the DNPH-containing derivatization solution. Proteins were blotted to a nitrocellulose membrane. After staining with red ponceau to quantifiy total proteins, carbonylated proteins were detected using the rabbit polyclonal anti-DNP primary antibody (1:100; Protein Oxyblot Kit) and the anti-rabbit peroxidase antibody. Detection was performed with ECL.

## siRNA transfection

SiUcp2 (Life technologies, ID:s75722), sip38a (Life technologies,ID:s77114), sip38γ (Life technologies, ID:s78078) and siCtrl (Life technologies, silencer negative control n°1) were transfected at 10 nm final concentration using lipofectamine RNAimax (Life technologies) from day 1 to day 3 or 4 of satellite cell culture.

## Statistical analysis

Statistical analysis was performed using GraphPad Prism software using the two-tailed unpaired t-test test and a minimum of 95% confidence interval for significance; $P$-values indicated on figures are <0.05 (*),<0.01(**) and <0.001 (***). Figures display mean values of all animals tested ±standard deviation (s.d), with all experiments being carried out on a minimum of 3 animals in 3 independent experiments.

# Acknowledgments

This work was supported by the Institut Pasteur, the CNRS (UMR 3738 and UMR 8256) and Sorbonne Universités, with grants to MB, DM and BF from the AFM, grants to MB and DM from the ANR (REGSAT), the Laboratoire d'Excellence REVIVE (Investissement d'Avenir ANR-10-LABX-73) and the EU Optistem project (Health, FP7-2007, 223098) and grant to AL from the DIM Stem Pôle. A L'honoré was supported by a postdoctoral fellowship from the Fondation pour la Recherche Médicale (FRM) and a Marie Curie IRG grant (MC-IRG248496/SATELLITE CELL). G Pallafacchina was supported by a postdoctoral fellowship from the Fondation pour la Recherche Médicale. The authors thank C Bodin, C Cimper, N Benanteur and the laboratory of histology of the Institut Pasteur for technical help, A Lombes and F Devaux for reagents and advice on mitochondrial function, R Blaise for reagents and B Robert for critical reading of the manuscript and discussion.

# Additional information

## Competing interests

Margaret Buckingham: Reviewing editor, *eLife*. The other authors declare that no competing interests exist.

## Funding

| Funder | Grant reference number | Author |
|---|---|---|
| Seventh Framework Programme | Marie Curie IRG 248496 | Aurore L'honoré |
| Fondation pour la Recherche Médicale | Postdoc Fellowship | Aurore L'honoré<br>Giorgia Pallafacchina |
| Domaine d'Intérêt Majeur STEM-Pole | | Aurore L'honoré |
| AFM-Téléthon | | Bertrand Friguet |
| Agence Nationale de la Recherche | REGSAT | Margaret Buckingham |
| Agence Nationale de la Recherche | ANR-10-LABX-73 | Margaret Buckingham |
| Seventh Framework Programme | OptiStem 223098 | Margaret Buckingham |
| AFM-Téléthon | | Didier Montarras |

The funders had no role in study design, data collection and interpretation, or the decision to submit the work for publication.

## Author contributions

Aurore L'honoré, Conceptualization, Data curation, Formal analysis, Investigation, Methodology, Writing—original draft, Project administration, Writing—review and editing; Pierre-Henri Commère, Data curation, Software, Formal analysis, Methodology; Elisa Negroni, Conceptualization, Resources, Methodology, Writing—review and editing; Giorgia Pallafacchina, Resources, Writing—review and editing; Bertrand Friguet, Jacques Drouin, Resources; Margaret Buckingham, Resources, Funding acquisition, Writing—original draft, Project administration, Writing—review and editing; Didier

Montarras, Conceptualization, Data curation, Supervision, Funding acquisition, Investigation, Writing—original draft, Project administration, Writing—review and editing

### Author ORCIDs
Aurore L'honoré http://orcid.org/0000-0001-6371-4455
Giorgia Pallafacchina https://orcid.org/0000-0001-9766-5970
Didier Montarras http://orcid.org/0000-0002-7691-359X

### Ethics

Animal experimentation: All animal procedures were approved and conducted in accordance with the Institut Pasteur animal ethics committee (CEEA Institut Pasteur n°2013-0017 and APAFIS #2455 2015 1122133311) following the regulations of the Ministry of Agriculture and the European Community guidelines. All surgery was performed under Ketamine/Xylazine anesthesia and every effort was made to minimize suffering.

### Decision letter and Author response

Decision letter https://doi.org/10.7554/eLife.32991.035
Author response https://doi.org/10.7554/eLife.32991.036

## Additional files

### Supplementary files

• Supplementary file 1. List of mouse lines and their use in the different experiments.
DOI: https://doi.org/10.7554/eLife.32991.029

• Supplementary file 2. List of mouse sequences used in qPCR. Key Resource Table.
DOI: https://doi.org/10.7554/eLife.32991.030

• Transparent reporting form
DOI: https://doi.org/10.7554/eLife.32991.031

### Data availability

All data generated or analysed during this study are included in the manuscript and supporting files. Source data files have been provided for figures and supplement figures.

The following previously published datasets were used:

| Author(s) | Year | Dataset title | Dataset URL | Database, license, and accessibility information |
|---|---|---|---|---|
| Giorgia Pallafacchina, Didier Montarras, Margaret Buckingham, B Regnault | 2010 | An adult tissue-specific stem cell in its niche: a gene profiling analysis of invivo quiescent and activated muscle satellite cells | https://www.ncbi.nlm.nih.gov/geo/query/acc.cgi?acc=GSE15155 | Publicly available at NCBI Gene Expression Omnibus (accession no: GSE15155) |

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
