## [Decision Letter]

Thank you for sending your article entitled "Redox regulation of muscle stem cells is critical for skeletal muscle regeneration" for peer review at *eLife*. Your article has been evaluated by three peer reviewers, and the evaluation has been overseen by a Reviewing Editor and Marianne Bronner as the Senior Editor.

Given the list of essential revisions, including new experiments, the editors and reviewers invite you to respond within the next two weeks with an action plan and timetable for the completion of the additional work. We plan to share your responses with the reviewers and then issue a binding recommendation.

Below we have compiled a summary of the essential revisions (please see attached full reviews for further details):

In the present manuscript by L'honore et al., the authors claim that *Pitx2* and *Pitx3* are involved in the regulation of the redox state in skeletal muscle stem cells, which balances their proliferation and differentiation for skeletal muscle regeneration. This manuscript follows from the authors earlier work, in which they identified *Pitx2* and *Pitx3* as regulators of redox state in fetal myogenesis. In the present work, the authors use single and double mutant *Pitx2/Pitx3* mice as well as conditional mutants to assess the role of *Pitx2/Pitx3* in adult satellite cell function related to ROS production. The authors conclude that *Pitx2/Pitx3* loss raises ROS levels promoting premature differentiation that can be rescued by reducing p38α signaling together with NAC supplementation. This work extends the group's prior work in fetal myogenesis to adult satellite cells revealing distinct phenotypes within the two contexts.

There are several major issues with this study:

1) The authors provide evidence linking the *Pitx2/Pitx3* expression levels and related amount of ROS production (redox state) with substantially different biological outcomes in adult satellite cells – either premature differentiation or senescence. The implication is that this dose relationship dictates whether a cell will differentiate or senesce. This needs to be tested more thoroughly. Does ROS activity change incrementally with *Pitx* gene dosage?

2) Manipulation of the redox state of muscle stem cells in vitro via combined p38α and ROS inhibition, the authors discover a ten-fold cell expansion and complete preservation of the regenerative capacity comparable to freshly isolated muscle stem cells. How is this possible? If the role of p38α is to regulate the redox state, shouldn't ROS inhibition on its own exert an effect of similar magnitude? Does NAC treatment alone improve grafting potential of cultured muscle stem cells? Resolving this interaction is important to resolve the mechanism.

3) Along the same line, one might wonder whether premature satellite cell differentiation by increased levels of ROS coincides with loss of asymmetric division. This can be evaluated by localization of pp38 and Par staining, as shown in Troy et al. (CSC 2012). In line with this thinking, it is possible that different SC subsets undergo different fates in response to *Pitx2/3* deletion. Can clonal analysis be performed to determine if SCs are biased to differentiate or senesce and is that prevented by p38i or NAC. It is possible that the synergistic effect of p38i and NAC occur because they are acting in different cells/states.

4) The authors should focus on clarifying whether these different outcomes depend entirely on p38α (regulating different subsets of genes depending on whether it is activated by moderate vs high ROS) or by the activation of parallel signalling by ROS (this seems to me the most plausible scenario, which would explain why SB/NAC treatment is more effective than each treatment alone).

The most effective way to do so would be to analyze gene expression (genome wide, if possible; or alternatively by measuring expression levels of muscle differentiation, cell cycle and senescence-associated genes by qPCR, and activation of alternative signalling pathways, such as NFkB, *JNK*, DDR etc., in satellite cells from *Pitx3^-/-^* mutant and *Pitx3^+/-^* control animals 3 days post cardiotoxin injection, with or without SB and NAC in different combinations).

5) It is important that the authors clarify the issue of whether *Pitx2/3* control only production of endogenous (mitochondrial) ROS, or also ROS generation in response to cytokines/growth factors, thereby clarifying the relationship between ROS- vs. cytokine/growth factor-mediated activation of p38 kinases α and β.

6) The effects of oxygen concentrations on satellite cell behavior are also well documented, first published in 2001, these studies demonstrate that un-physiological O_2_ concentrations (20%) promote myogenic differentiation and repress proliferation. There is a concern that all the culture experiments are done in a hyperoxic environment. Thus, the NAC treatment and other ROS manipulation in culture are potentially the same that would be achieved by simply reducing O_2_. The authors should test whether the *Pitx2/3* mutants have ROS effects that are rescued by NAC in 4-6% O_2_ culture. Ignoring the effects of O_2_ culture levels considering the extensive published literature is a serious concern.

7) The transplantation experiment does not address whether the treatment enhances satellite cell expansion but simply shows that more dystrophin+ myofibers are present when treated cells are transplanted vs. non-treated cells. This could reflect either (i) increased survival upon transplantation, (ii) increased differentiation, (iii) a combination of (i) and (ii), or (iv) expansion of the satellite cells. The authors fail to show whether more donor satellite cells are present and thus the author's claims in the discussion are not supported by the data. The authors should perform transplants and double injuries in presence or absence of treatments followed by quantification of muscle fiber and SC contribution.

8) Many of the controls are *Pitx2/Pitx3* heterozygotes and there is a concern that further manipulation of cells could uncover haploinsufficiency. The authors need to include wild type controls for some of the experiments as differing experimental conditions such as knock downs or changing culture conditions could uncover a haploinsufficiency phenotype that is not observed under normal culture conditions.

9) The title and narrative of the manuscript imply that the authors directly regulate ROS. The authors should reframe the title and the manuscript in general, to refocus on the role of *Pitx2/3* in adult muscle stem cells. Alternatively, the authors need to directly manipulate mitochondrial function in vivo.

Reviewer #1:

In the present manuscript by L'honore et al., the authors claim that *Pitx2* and *Pitx3* are involved in the regulation of the redox state in skeletal muscle stem cells, which balances their proliferation and differentiation for skeletal muscle regeneration. The authors additionally claim that manipulation of the muscle stem cell redox state can expand muscle stem cells in vitro and preserve their regenerative capacity similar to freshly isolated cells. These claims are well supported by extensive temporally-controlled conditional mouse genetics and in vitro muscle stem cell culture experiments, and the body of work presents an important new addition to the muscle stem cell literature.

The finding of *Pitx2/3*'s role in regulating the muscle stem cell redox state is certainly not surprising given the author's previous publication on the role of *Pitx2/3* in fetal myogenic progenitor cells. Yet, the extension of that work to the adult muscle stem cell is appreciated as the adult muscle stem cell becomes activated from a quiescent state in adult muscle regeneration, which differs from the highly proliferative muscle progenitor population, which is continuously cycling.

Manipulating the redox state of muscle stem cells in vitro via combined p38α and ROS inhibition, the authors discover a ten-fold cell expansion and complete preservation of the regenerative capacity comparable to freshly isolated muscle stem cells. While the data is very exciting, the rationale for implementing the dual inactivation strategy is lacking and the results need a much more in-depth discussion within the context of the vast p38α literature on muscle stem cells. In particular, the magnitude of the cell expansion effect by dual p38α and ROS inhibition is not simply cumulative but much more substantive. How is this possible? If the role of p38α is to regulate the redox state, shouldn't ROS inhibition on its own exert an effect of similar magnitude? Along this line, it would be helpful to expand the in vitro inhibition (and transplantation) studies to include a dose response for SB203580, for which partial inhibition of p38α was shown to rejuvenate aged muscle stem cells (Bernet, 2014). Would NAC treatment alone improve grafting potential of cultured muscle stem cells? Minimally the results need a much more in-depth discussion including work from the Olwin lab (Jones, 2005; Bernet, 2014), Blau lab (Cosgrove, 2014), and in particular, work from the Pell lab (Brien, 2013), which used mouse genetics to completely eliminate p38α from the muscle stem cell.

Reviewer #2:

The authors use single and double mutant *Pitx2/Pitx3* mice as well as conditional mutants to assess the role of *Pitx2/Pitx3* in satellite cell function related to ROS production. The authors conclude that *Pitx2/Pitx3* loss raises ROS levels promoting premature differentiation that can be rescued by reducing p38α signaling together with NAC supplementation.

The results, while of interest appear an indirect method to assess the effects of ROS on satellite cell differentiation. Although the data clearly show that ROS levels are changing, it seems likely that loss of *Pitx2/Pitx3* affects far more than ROS levels and thus, the authors cannot simply attribute the deficits in muscle regeneration occurring upon loss of *Pitx2/Pitx3* to changes in ROS. There are a number of serious concerns regarding the manuscript that are discussed in the following paragraphs but the overall issue that cannot be overcome is the use of *Pitx2/Pitx3* knockouts to study ROS in satellite cells. The effects on ROS are indirect and thus, how can the authors attribute the effects of the double knockout to changes in ROS levels? Floxed alleles are available for both *Ucp2* and *Tfam* from Jax. Incorporation of these knockouts in satellite cells and a comparison with *Pitx2/Pitx3* knockouts is essential for the authors to draw conclusions about ROS effects on satellite cell behavior and separate the non-ROS dependent effects of *Pitx2/Pitx3* knock outs from ROS-dependent effects.

Additional major issues:

The authors discussion and depiction for the roles of p38α MAPK signaling are insufficient and dated. Several laboratories have demonstrated that p38α plays a role in satellite cell activation and asymmetric division. The subcellular localization of p38α is a critical determinant of its function and during satellite cell activation/proliferation the phospho- p38α is present in the cytoplasm. The role for p38α in differentiation is well-documented and is nuclear. The authors are likely partially inhibiting p38α using the SB203580 concentrations stated and this is well documented by several labs to dramatically enhance transplantation efficacy. Thus, the p38α effects observed by the authors do not add to what has already been published.

The effects of oxygen concentrations on satellite cell behavior are also well documented, first published in 2001, these studies demonstrate that un-physiological O_2_ concentrations (20%) promote myogenic differentiation and repress proliferation. Reducing the O_2_ concentrations to physiological levels of 2-6% would likely produce the same effect as NAC in the manuscript. Ignoring the effects of O_2_ culture levels in light of the extensive published literature is a serious concern and needs to be addressed by the authors.

The observations that NAC/SB treatment expands the satellite cell population is predicted from the effects of O_2_ concentrations and the effects of p38α inhibition on satellite cell self-renewal and transplantation. Thus, this observation is not novel and does not add significantly to existing knowledge. The transplantation experiment does not address whether the treatment enhances satellite cell expansion but simply shows that more dystrophin+ myofibers are present when treated cells are transplanted vs. non-treated cells. This could reflect either (i) increased survival upon transplantation, (ii) increased differentiation, (iii) a combination of (i) and (ii), or (iv) expansion of the satellite cells. The authors fail to show whether more donor satellite cells are present and thus the claim at the end of the Results "To test whether engrafted cells also contributed to the satellite cell compartment in recipient muscles, we performed immunostaining with GFP and *Pax7* antibodies. Both staining co-localized in a subset of nuclei in the engrafted muscle, indicating that the transplanted cells also successfully populated the satellite cell compartment (Figure 6I)" and claims in the Discussion are not supported by the data.

The vast majority of the controls are *Pitx2/Pitx3* heterozygotes and there is a concern that further manipulation of cells could uncover haploinsufficienies. The authors need to include wild type controls for the majority of the experiments as differing experimental conditions such as knock downs or changing culture conditions could uncover a haploinsufficiency phenotype that is not observed under normal culture conditions.

Finally, the global knockout data reported in Figures 1-3 need repeating with the floxed allele as the authors cannot attribute the effects observed as cell intrinsic. The effects observed in vivo could be due to a combination of effects on multiple cell types, especially those shown in vivo. The authors imply that the global knockout and the satellite cell-specific knockouts are equivalent but they are not. There seems to be no rationale for performing half of the experiments on a global knock out and the other half on a cell-specific knock out.

In summary, the authors utilize an indirect approach to assess the effects of changing ROS levels in satellite cells. The possibility that the observed phenotypes are due to effects independent of ROS are not sufficiently addressed. Moreover, the mixed use of global and cell-specific knock outs complicates the interpretation of the phenotypes. More direct analyses of ROS effects on satellite cells could be performed and coupled with data from the submitted manuscript to much better support the conclusions drawn by the authors.

Reviewer #3:

This is a very interesting manuscript, in which the authors followed-up previous studies that identified *Pitx2* and *Pitx3* as satellite cell regulators of redox state, which in turn regulates DNA damage and differentiation. By using *Pitx2/3* single and double mutants, as genetic models of perturbed redox states in satellite cells, the authors show that while moderate ROS production (at physiological levels) leads to premature satellite cell differentiation, via cell cycle entry and p38 activation, excessive ROS levels induce senescence, DNA damage and consequent activation of the differentiation checkpoint which inhibits satellite cell differentiation. The authors used this information to devise a combinatorial and transient blockade of ROS and p38α kinase to expand satellite cells for successful muscle engraftment in vivo, upon transplantation.

Overall, the experiments are well performed and the data correctly interpreted, revealing an interesting connection between genetic control of redox state by *Pitx2/3* and p38 activation in satellite cells, leading to physiological or premature activation of the myogenic program, or DNA damage, cellular senescence and block of differentiation.

Below are listed few important points that I invite the authors to address.

1) In Figure 1, the gene expression analysis does not include satellite cells from injured muscles. That experimental point should be included, as it seems it would provide an important dataset integration.

2) It should be explained why the authors initially used pectoralis, diaphragm and abdominal muscles, and notexin injury (Figure 1), and then switched to hind-limb muscles and cardiotoxin injury for satellite cell analysis.

3) Figure 1J. The ROS levels in satellite cells from control and mutant mice seem to eventually converge at later time points (D3). The authors should comment on this trend and possibly extend their analysis beyond D3 (i.e. D4 and 5).

4) ROS-mediated activation of p38 is an interesting finding, that adds one more p38 activator to stimuli identified so far (cytokines, cell to cell contact). It would be interesting to experimentally address (or just speculate/discuss) the relationship between these stimuli and intracellular production of ROS. For instance, is TNFα-mediated activation of p38 in satellite cells affected by NAC?

5) Along the same line, one might wonder whether premature satellite cell differentiation by increased levels of ROS actually coincides with loss of asymmetric division. This can be evaluated by localization of p-p38 and Par staining, as shown in Troy et al. (CSC 2012).

6) Figure 3I. The reduction of p38α phosphorylation upon incubation with the p38 kinase inhibitor SB is unexpected, as the inhibitor should not affect p38 phosphorylation (which is catalyzed by upstream kinases – MKK3, MKK6 and others). Actually, a paradoxical hyper-phosphorylation should be observed, as a result of the accumulation of phosphorylated p38 upon blockade by SB. The authors might want to re-evaluate this experiments (possibly by showing also the total p38 Western blot).

7) *Pax3*/GFP sorting of satellite cells is definitely acceptable; however, validation of the major findings (ROS-p38 activation, and expansion by NAC/SB ex vivo) should be done in satellite cells sorted by conventional FACS procedures.

8) Improved engraftment of satellite cells injected after in vitro expansion by NAC/SB treatment should be evaluated also upon transplantation in muscles of wild type animals and repeated injuries, in order to evaluate whether these cells divide asymmetrically – a key issue in regenerative medicine.

9) A similar finding reporting on DNA damage-driving senescence that impairs satellite cell differentiation has been recently reported by Latella et al. and should be discussed.

[Editors' note: the authors’ plan for revisions was approved and they made a formal revised submission. What follows is both their initial plan and final response submitted with their revised article]

---

## [Author Response]

[…] 1) The authors provide evidence linking the Pitx2/Pitx3 expression levels and related amount of ROS production (redox state) with substantially different biological outcomes in adult satellite cells – either premature differentiation or senescence. The implication is that this dose relationship dictates whether a cell will differentiate or senesce. This needs to be tested more thoroughly. Does ROS activity change incrementally with Pitx gene dosage?

Revision plan: We think that our data in Figure 1J indicate that ROS levels change incrementally with *Pitx* gene dosage. While in the *Pitx3* mutant, we observe a moderate increase of ROS compared to control both at days 2 and 3, a higher increase is found in the double mutant. The *Pitx2* mutant alone does not show a significant increase in ROS levels compared to the control. To be more clear in the manuscript, we proposed to present these data in a histogram, instead of linear plots.

As noted by reviewer 3, we chose not to do ROS quantification at days 4 and 5 in the mutants in order to focus on the role of ROS in early differentiation, marked by down-regulation of *Pax* genes, and up-regulation of *Myogenin* (Figures 1E, G).

Final response: The data are now presented in a histogram (Figure 1J).

*2) Manipulation of the redox state of muscle stem cells in vitro via combined p38*α *and ROS inhibition, the authors discover a ten-fold cell expansion and complete preservation of the regenerative capacity comparable to freshly isolated muscle stem cells. How is this possible? If the role of p38*α *is to regulate the redox state, shouldn't ROS inhibition on its own exert an effect of similar magnitude? Does NAC treatment alone improve grafting potential of cultured muscle stem cells? Resolving this interaction is important to resolve the mechanism.*

Revision plan: As indicated in the manuscript, the use of SB and NAC alone are not sufficient to prevent complete p38α activation, giving a first explanation for the better expansion of satellite cells in the presence of both compounds. While it has been previously shown that p38α can directly regulate the redox state in some cells, such as hematopoietic stem cells, in muscle stem cells, this does not appear to be the case, as shown in Figure 6—figure supplement 1E, where SB treatment alone does not significantly affect ROS levels. We propose that in muscle stem cells, p38α does not act as a redox regulator, but rather as a redox signalling molecule.

To address the issue of the role of NAC on grafting potential of cultured muscle stem cells, we proposed to perform the following grafting experiments: satellite cells purified from *Pax3*(GFP) cells will be cultured in the absence or presence of NAC, and then grafted in the *Tibialis Anterior (TA*) muscle of *Rag2^-/-^^-/-^:Il2Rβ^-/-^:dmd/dmd* mice. Three weeks after grafting, the grafting efficiency will be analyzed by detection of dystrophin in regenerating fibers as in our previous experiments.

Final response: These additional grafting experiments are now presented in Figure 6H-K. We have moved Figure 6F (schema) and Figure 6I (*Pax7* and *Pax3^GFP^* images) into Figure 6—figure supplement 1I and J.

The question of NAC versus SB effects are now addressed more extensively in the Discussion.

3) Along the same line, one might wonder whether premature satellite cell differentiation by increased levels of ROS coincides with loss of asymmetric division. This can be evaluated by localization of pp38 and Par staining, as shown in Troy et al. (CSC 2012). In line with this thinking, it is possible that different SC subsets undergo different fates in response to Pitx2/3 deletion. Can clonal analysis be performed to determine if SCs are biased to differentiate or senesce and is that prevented by p38i or NAC. It is possible that the synergistic effect of p38i and NAC occur because they are acting in different cells/states.

Revision plan: To address the issue of a possible link between premature satellite cell differentiation and loss of asymmetric division, we will study the distribution of pp38α and Par in cultured control and *Pitx3^-/-^* satellite cells. It is not clear whether clonal analysis could provide more information than we already have from the mass culture studies. Pp38 staining will not help to distinguish cells undergoing differentiation from cells undergoing senescence since both will be positive. However, we propose to add to Figure 5F-H, data obtained by treating the *Pitx2/3* double mutant cells by both NAC and SB.

Final response: We could detect asymmetric distribution of pp38α and Par in cultured control and *Pitx3^-/-^* satellite cells. However the results on their precise localisation were not statistically significant with considerable variation over the cell population. We used the same sources of antibodies as those reported by Troy et al. but unfortunately in our hands the subcellular resolution was poor.

We now include data that address the fate of the *Pitx* mutant cells, with addition of SB and NAC together as well as separately (Figure 5E, Figure 6I-K). These results provide additional insight into the synergistic effect.

As now shown by additional experiments and discussed at greater length in the paper we think that NAC and therefore the level of ROS has additional effects, notably on DNA damage, senescence and cloning efficiency for example, which do not depend on P38α and are not targeted by SB.

*4) The authors should focus on clarifying whether these different outcomes depend entirely on* p38α *(regulating different subsets of genes depending on whether it is activated by moderate vs high ROS) or by the activation of parallel signalling by ROS (this seems to me the most plausible scenario, which would explain why SB/NAC treatment is more effective than each treatment alone).*

The most effective way to do so would be to analyze gene expression (genome wide, if possible; or alternatively by measuring expression levels of muscle differentiation, cell cycle and senescence-associated genes by qPCR, and activation of alternative signalling pathways, such as NFkB, JNK, DDR etc., in satellite cells from Pitx3^-/-^ mutant and Pitx3^+/-^ control animals 3 days post cardiotoxin injection, with or without SB and NAC in different combinations).

Revision plan: We agree that it is plausible that ROS can act through parallel signalling to p38α. To test this hypothesis, we propose to perform Western-blot analysis on control and *Pitx3^-/-^* satellite cells cultured with and without SB and NAC in different combinations. We think that RNA transcripts are less informative about the activation of signalling pathways than looking at the phosphorylated protein intermediates. We therefore propose to perform the following Western blots: phosphoATM/ATM (DDR), phosphoJun/Jun (*JNK*), phosphop65/p65 (NFkB).

Final response: We have carried out Western blots for phosphorylated versus unphosphorylated *JNK* and ERK which are more informative than p65 (NFkB) or ATM (DDR), neither of which changed at the onset of differentiation. Representative Western blots and quantification are shown in Figure 3—figure supplement 1H and 1I. We do not detect significant changes in *Pitx3* mutant satellite cells, indicating that these pathways are not affected.

We have modified the summary diagram (Figure 7) to indicate that ROS may act independently of p38α signalling. We think that DNA damage is a clear example of this as we show and is now pointed out in the discussion where other effects of ROS, on *Sirt1* for example, are also invoked.

5) It is important that the authors clarify the issue of whether Pitx2/3 control only production of endogenous (mitochondrial) ROS, or also ROS generation in response to cytokines/growth factors, thereby clarifying the relationship between ROS- vs. cytokine/growth factor-mediated activation of p38 kinases α and β.

Revision plan: Our previous (L’honore et al., 2014) and current results implicate *Pitx2/3* transcription factors as regulators of mitochondrial ROS production through regulation of mitochondrial activity. To investigate whether they can also control ROS generation in response to cytokines, we propose to culture control and *Pitx3* mutant cells in the absence or presence of TNFα (suggested by reviewer 3) and NAC. Differentiation and ROS levels will be monitored. If *Pitx3* control ROS production by TNFα stimulus, TNFα should have no effect in mutant cells, while affecting control cells.

Final response: The experiment with TNFα is shown in Figure 3—figure supplement 1J. The result shows that mutant cells are sensitive to a similar extent to control cells, in keeping with a role of *Pitx* factors in regulating mitochondrial ROS independently of cytokines. Furthermore, in wild type cells, we show that TNFα increases ROS levels and affects the balance between proliferation and differentiation (Figure 3J).

*6) The effects of oxygen concentrations on satellite cell behavior are also well documented, first published in 2001, these studies demonstrate that un-physiological* O_2_*concentrations (20%) promote myogenic differentiation and repress proliferation. There is a concern that all the culture experiments are done in a hyperoxic environment. Thus, the NAC treatment and other ROS manipulation in culture are potentially the same that would be achieved by simply reducing* O_2_*. The authors should test whether the Pitx2/3 mutants have ROS effects that are rescued by NAC in 4-6%* O_2_*culture. Ignoring the effects of* O_2_*culture levels considering the extensive published literature is a serious concern.*

Revision plan: We agree with reviewer 2 that the effects of O_2_ culture levels on cells is a serious concern. While a hyperoxic environment has been previously shown to negatively affect myogenic proliferation, the role of intracellular ROS levels in this phenotype has not been established. In contrast, we have previously demonstrated that culturing fetal muscle cells under 20% or 3% O_2_ (L’honore et al., 2014) does not change their ROS levels, and importantly does not affect their response to NAC treatment. In this context, we have already obtained evidence that the phenotype of *Pitx2/3* double mutant cells is not rescued by culture in 3% O_2._ We propose to add these results to the new manuscript and to perform satellite cell culture in the absence or presence of NAC under hyperoxic (20% O_2_) and normoxic conditions (3% O_2_).

Final response: Results of culture in 3% oxygen have now been added as Figure 3—figure supplement 1G, Figure 5—figure supplement 1G and Figure 6—figure supplement 1B. We show that the effect of NAC and therefore modulation of ROS levels on the *Pitx3* mutant phenotype is not rescued by satellite cell culture under these normoxic conditions (Figure 3—figure supplement 1G). This is also the case for double *Pitx2/3* mutant cells (Figure 5—figure supplement 1G) and for wild-type cells (Figure 6—figure supplement 1B).

7) The transplantation experiment does not address whether the treatment enhances satellite cell expansion but simply shows that more dystrophin+ myofibers are present when treated cells are transplanted vs. non-treated cells. This could reflect either (i) increased survival upon transplantation, (ii) increased differentiation, (iii) a combination of (i) and (ii), or (iv) expansion of the satellite cells. The authors fail to show whether more donor satellite cells are present and thus the author's claims in the discussion are not supported by the data. The authors should perform transplants and double injuries in presence or absence of treatments followed by quantification of muscle fiber and SC contribution.

Revision plan: The issue of the re-colonisation of the satellite cell reservoir will be addressed by grafting experiments under the following conditions: satellite cells purified as *Pax3*(GFP) cells will be cultured in the absence or presence of NAC, SB and NAC/SB and will then be grafted into the *TA* muscles of either *Rag2^-/-^:Il2Rβ^-/-^or Rag^-/-^:Il2Rβ^-/-^:dmd/dmd* mice. Three weeks after grafting, the presence of GFP+/*Pax7+* satellite cells will be quantified.

We propose to perform a single injury experiment and not a double one that will be too time consuming and will not add much information.

Final response: This grafting experiment with both NAC and SB as well as each separately is now presented in Figure 6H-K. It shows an additional increase in regeneration (Figure 6J, K) in the presence of both with a striking increase in the expansion of grafted *Pax3*(GFP)-positive cells after injury (Figure 6I).

8) Many of the controls are Pitx2/Pitx3 heterozygotes and there is a concern that further manipulation of cells could uncover haploinsufficiency. The authors need to include wild type controls for some of the experiments as differing experimental conditions such as knock downs or changing culture conditions could uncover a haploinsufficiency phenotype that is not observed under normal culture conditions.

Revision plan: We have evidence that results obtained with cells isolated from wild type *Pax3^GFP/+^*mice are not statistically different to those obtained with cells from *Pitx2^+/-^ :Pitx3^+/-^ : Pax3^GFP/+^* mice, suggesting absence of haploinsufficiency. These results will be included in the new manuscript.

Final response: The issue of haploinsufficiency is addressed by data on wild type and heterozygote satellite cells, which show similar levels of ROS (Figure 1—figure supplement 1M).

*9) The title and narrative of the manuscript imply that the authors directly regulate ROS. The authors should reframe the title and the manuscript in general, to refocus on the role of Pitx2/3 in adult muscle stem cells. Alternatively, the authors need to directly manipulate mitochondrial function* in vivo.

Revision plan: We will change the title of the manuscript in order to refocus on the role of *Pitx2/3* in adult muscle stem cells. We will also include new references on p38 in our Discussion.

Final response: The title has been modified. The manuscript begins with the results on the *Pitx* mutants and this is now the starting point for the Discussion, which now further addresses the points raised by the reviewers.

Additional references have been added.

We have not removed any of the previous results presented but some of these have now been transferred to the supplementary figures to make room for the most important new data obtained following the reviewers’ recommendations. We thank them for suggestions that have strengthened the manuscript.